# Magneto-active elastic shells with tunable buckling strength

Dong Yan[1], Matteo Pezzulla[1], Lilian Cruveiller[1,2], Arefeh Abbasi[1] & Pedro M. Reis [1]✉

Shell buckling is central in many biological structures and advanced functional materials, even if, traditionally, this elastic instability has been regarded as a catastrophic phenomenon to be avoided for engineering structures. Either way, predicting critical buckling conditions remains a long-standing challenge. The subcritical nature of shell buckling imparts extreme sensitivity to material and geometric imperfections. Consequently, measured critical loads are inevitably lower than classic theoretical predictions. Here, we present a robust mechanism to dynamically tune the buckling strength of shells, exploiting the coupling between mechanics and magnetism. Our experiments on pressurized spherical shells made of a hard-magnetic elastomer demonstrate the tunability of their buckling pressure via magnetic actuation. We develop a theoretical model for thin magnetic elastic shells, which rationalizes the underlying mechanism, in excellent agreement with experiments. A dimensionless magneto-elastic buckling number is recognized as the key governing parameter, combining the geometric, mechanical, and magnetic properties of the system.

[1] Flexible Structures Laboratory, Institute of Mechanical Engineering, École Polytechnique Fédérale de Lausanne (EPFL), Vaud, Switzerland. [2] École Polytechnique, Palaiseau, France. ✉email: pedro.reis@epfl.ch

Shells are curved thin structures that can withstand extreme loading conditions due to the interplay between bending and stretching deformation[1,2] through the so-called shell effect[3]. Thin shells are a ubiquitous structural element in engineering[4] and also widely observed in nature across length scales, from viruses[5], and capsules[6–8], to pollen grains[9], and plants[10]. While curvature is the key ingredient underlying the excellent mechanical performance of shells, it is also responsible for the catastrophic (subcritical) nature of their elastic instabilities[11]. Consequently, shells are high sensitivity to imperfections[1,12–14]. For over a century, the pressure buckling of a spherical shell has been a long-standing canonical problem of elastic (in)stability[1,2]. When the in-out pressure differential across a shell exceeds a threshold value, the shell loses its load-carrying capacity[1]. The ensuing collapse is unpredictable, occurring at loads that are significantly lower than classic theoretical predictions[14]. Consequently, measuring and predicting the critical buckling pressure has proven to be nontrivial due to the high imperfection sensitivity.

In 1915, Zoelly[15] derived the theoretical prediction for the critical buckling pressure of a perfect spherical shell, of radius $R$ and thickness $h$, loaded under a uniform pressure $p$:

$$p_c = \frac{2E}{\sqrt{3(1-\nu^2)}} \left(\frac{R}{h}\right)^{-2}, \tag{1}$$

where $E$ and $\nu$ are the Young's modulus and Poisson's ratio of the material, respectively. Notwithstanding this classic result, given the high imperfection sensitivity, buckling pressures measured in experiments, $p_{max}$, have long been found[1,12–14] to be much lower than the prediction from Eq. (1). This discrepancy generated a long debate in the shell mechanics community that lasted for nearly four decades until the disagreement between theory and experiments was finally attributed to imperfections[16]. Given this intrinsic mismatch, it became standard to define an empirical quantity, the so-called knockdown factor,

$$\kappa_d = \frac{p_{max}}{p_c}, \tag{2}$$

which is always smaller than unity, spreading a wide range[14,17]: $0.05 \leq \kappa_d \leq 0.9$.

In engineering, significant efforts have focused on improving predictions for the knockdown factor and understanding how it is affected by imperfections[1,12–14]. A breakthrough came only recently, from experiments, when the combination of a rapid prototyping technique for spherical shells[18] and its adaptation to seed precisely designed defects led to a quantitative relationship between the knockdown factor and defect geometry[17]. This study demonstrated that if imperfections can be measured precisely, then the knockdown factor can be precisely predicted using an appropriate shell theory[19–25], thus opening the door for less conservative designs of shell structures. In parallel, nondestructive probing methods have recently been proposed to experimentally access the stability landscape of shells[26–28], while accepting the inevitable presence of multiple imperfections to set the knockdown factor. Furthermore, bilayer shells undergoing differential swelling were recently shown to have a varying knockdown factor during a transient, demonstrating how the knockdown factor can be modified post-fabrication, albeit not reversibly on-demand[29]. Still, to date, the knockdown factor is regarded as an intrinsic structural property dictated by imperfections imparted during fabrication or encoded at the design stage.

To liberate the predestination of imperfections, a shell structure requires a mechanism that can provide active control over buckling; the magneto-elastic coupling offers an opportunity to achieve this. In the literature, mechanical systems built comprising magneto-active materials, due to their active response to external magnetic fields, have been used in various applications for sensing[30], actuation[31,32], and control[32,33]. Past pioneering studies also addressed the deformation and buckling of magneto-elastic structures made of superparamagnetic[34–40] or soft-ferromagnetic[41–47] materials under an external magnetic field. More recent advances have tended to focus on hard-magnetic soft materials[48–51], which are magnetorheological elastomers (MREs) embedded with hard-ferromagnetic particles. This class of materials is magnetically hard with programmable remanent magnetization and high coercivity upon saturation, but mechanically compliant due to the soft elastomeric matrix. These characteristics enable fast, reversible, and complex shape-morphing through remote magnetic actuation, as has been exploited for functional devices in a variety of applications, including shape-programmable materials[48,49], biomedical devices[33], and soft robots[52–54].

Here, we propose a robust mechanism to dynamically tune the buckling strength of shells (i.e., their knockdown factor), by coupling elastic deformation and magnetic actuation. Our framework leverages the unique features of hard-magnetic soft materials in the context of shell mechanics. Specifically, we manufacture spherical shells made of a hard-magnetic elastomer (hereon referred to as magnetic shells) and characterize their critical buckling pressure under an applied magnetic field. We demonstrate that the knockdown factor can be modified externally by adjusting the magnitude and polarity of the field. To rationalize our results, we develop an axisymmetric magnetic shell theory, providing a physical interpretation of the interplay between shell mechanics and magnetic fields. Furthermore, we uncover the magneto-elastic buckling number (a dimensionless quantity that combines the magnetic, elastic, and geometric properties), which acts as the single governing parameter of the system. Finally, we provide evidence through experiments and finite element simulations that our tunable mechanism is general and extends to non-axisymmetric conditions when the magnetization and/or the applied field are not aligned with the defect.

## Results

**Tuning the knockdown factor of pressurized magnetic shells.** In our experiments, we position a hemispherical shell of radius, $R = 25.4$ mm, and thickness, $h = 321.2$ μm, in between a set of Helmholtz coils (Fig. 1a, b). These two coils impose a steady, uniaxial, and uniform magnetic field (flux density vector $\mathbf{B}^a = B^a\mathbf{e}_3$) on the shell, perpendicularly to its equatorial plane (see Methods section and Supplementary Notes 1 and 2 for details). The shell is made of a hard-magnetic MRE, a composite of NdPrFeB particles (volume fraction 7%) and vinylpolysiloxane (VPS) polymer (see Methods). The full 3D configuration and the corresponding cross-section profile of the shell are visualized through X-ray micro-computed tomography (μCT, 100 Scanco Medical AG), a representative example of which is presented in Fig. 1c. To measure the buckling strength, we depressurize the shell using a pneumatic-loading system under imposed-volume conditions and measure the associated pressure sustained by the shell (Methods section and Supplementary Note 2). Figure 1d presents the load-carrying behavior of the shell, characterized by the pressure ($p$) as a function of the volume change ($\Delta V$), both of which are normalized, respectively, by the classic buckling prediction, $p_c$, and the corresponding volume change immediately prior to buckling[19,20], $\Delta V_c = 2\pi(1-\nu)R^2h/\sqrt{3(1-\nu^2)}$. The onset of buckling corresponds to the maximum of each curve, $\overline{p}_{max} = p_{max}/p_c$, and the accompanying pressure drop indicates the loss of load-carrying capacity of the shell.

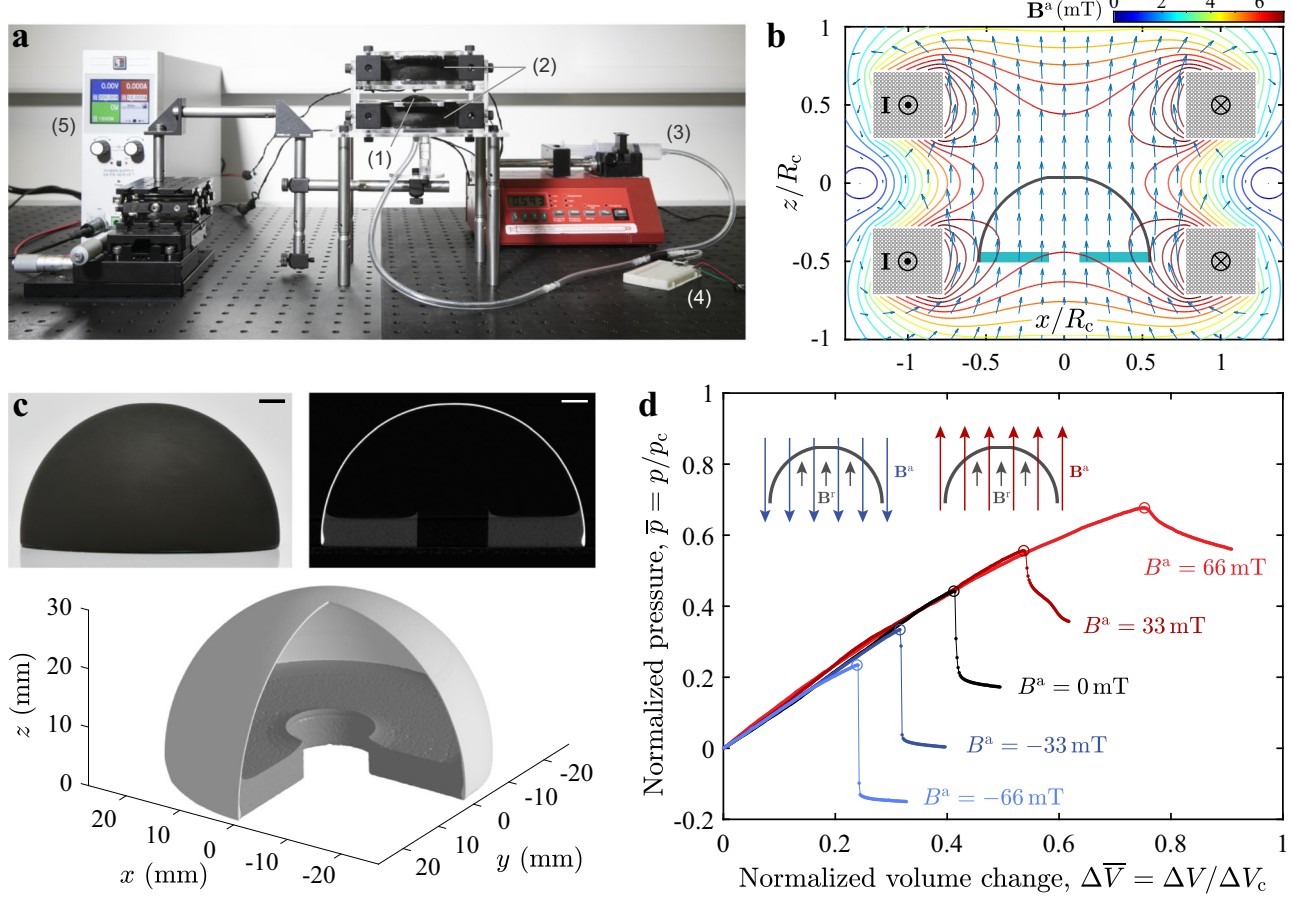

**Fig. 1 Buckling of a magnetic shell under combined pressure and magnetic loading. a** Photograph of the experimental apparatus: (1) a magnetic shell is positioned in between (2) a set of Helmholtz coils and depressurized using a pneumatic-loading system, comprising (3) a syringe pump and (4) a pressure sensor. The Helmholtz coils are driven by the current $\mathbf{I}$ in the direction shown in **b**, output from (5) a DC power supply . **b** Computed field of the vector (arrow) and magnitude (contour line) of the magnetic flux density $\hat{\mathbf{B}}^a$ generated by the coils, under current $\|\mathbf{I}\| = 1\,A$ (Supplementary Note 2). The field is uniaxial and uniform in the central region with a flux density of $\mathbf{B}^a = B^a\mathbf{e}_3$. Given axisymmetry, the field is represented in the $x$–$z$ plane ($y = 0$). Coordinates $x$ and $z$ are normalized by the coil radius $R_c = 46.5$ mm. **c** Photograph of the representative imperfect shell specimen (top left). Image of its cross-section (top right) scanned through x-ray $\mu$CT and the reconstructed 3D view (bottom, one quarter was artificially hidden to aid with visualization). Scale bars are 5 mm. **d** Loading curves of pressure, $\overline{p} = p/p_c$, versus volume change, $\Delta\overline{V} = \Delta V/\Delta V_c$, for the shell shown in **c**. The peak value represented by the open circle on each curve is the critical buckling pressure. The buckling test is performed at different levels of external flux density $B^a$. Inset: schematic of a shell subjected to a magnetic field $\mathbf{B}^a$, which is opposite (left) or parallel (right) to the shell magnetization $\mathbf{B}^r$.

In the absence of magnetic field ($B^a = 0$ mT), the representative shell shown in Fig. 1c buckles at the dimensionless pressure level of $\overline{p}_{max} = 0.44$. This value is far below the classical prediction of 1 ($p_{max} = p_c$), the reason being that we intentionally seed a precisely engineered dimple-like defect at the pole during manufacturing (Methods section and Supplementary Note 1), so as to consider the influence of imperfections in a controllable manner. Hence, the geometry of the shell deviates slightly from a perfect hemisphere by an amplitude of $\delta$ over the region of polar angle $0 \leq \varphi \leq \varphi_\circ$. The half angular width of the defect is then measured by $\varphi_\circ$, usually rescaled as $\overline{\varphi}_\circ = \varphi_\circ(\sqrt{12(1-\nu^2)}R/h)^{\frac{1}{2}}$[17,19,21,25]. For this test shell, the defect geometry is characterized as $\overline{\delta} = \delta/h = 0.39$ and $\varphi_\circ = 11.7°$ ($\overline{\varphi}_\circ = 3.2$) using an optical profilometer (Methods section and Supplementary Note 1). Although the introduced defect is too small to be seen with the naked eye, the buckling strength of the shell is dramatically reduced, by more than a half due to the high sensitivity to imperfections[1,17,19].

We proceed by focusing on the effect of the magnetic field on the buckling instability. The magnetized shell possesses a residual flux density of $\mathbf{B}^r = B^r\mathbf{e}_3$ ($B^r = 63$ mT, see Methods section and

Supplementary Note 1), and is responsive to an external magnetic field, with a significant modification of the buckling pressure (Fig. 1d) compared to the no-field case. When the field vector is parallel to the axis of magnetization (i.e., $\mathbf{B}^a$ and $\mathbf{B}^r$ are in the same direction), we observe an increase of the critical load by 12% and 24% under the flux densities $B^a = 33$ mT and $B^a = 66$ mT, respectively. Meanwhile, the accompanying pressure drop at the onset of buckling is gradually reduced, eventually disappearing for $B^a = 66$ mT. Therefore, the shell is strengthened by the applied field, and the buckling event becomes less catastrophic. By contrast, for the opposite field polarity (i.e., $\mathbf{B}^a$ and $\mathbf{B}^r$ are in opposite directions), the shell is weakened; the critical load decreases with a consequent more abrupt pressure drop past the buckling event ($B^a = \{-33, -66\}$ mT in Fig. 1d). These findings demonstrate that the intrinsic buckling strength of a magnetically active shell can be modified (increased or decreased), under an external magnetic field, on-demand.

To further explore the effect of magnetism on the buckling strength of pressurized spherical shells, we test shells with different defect geometries, over a range of external flux densities. As shown in the photographs of the specimens in Fig. 2a, the

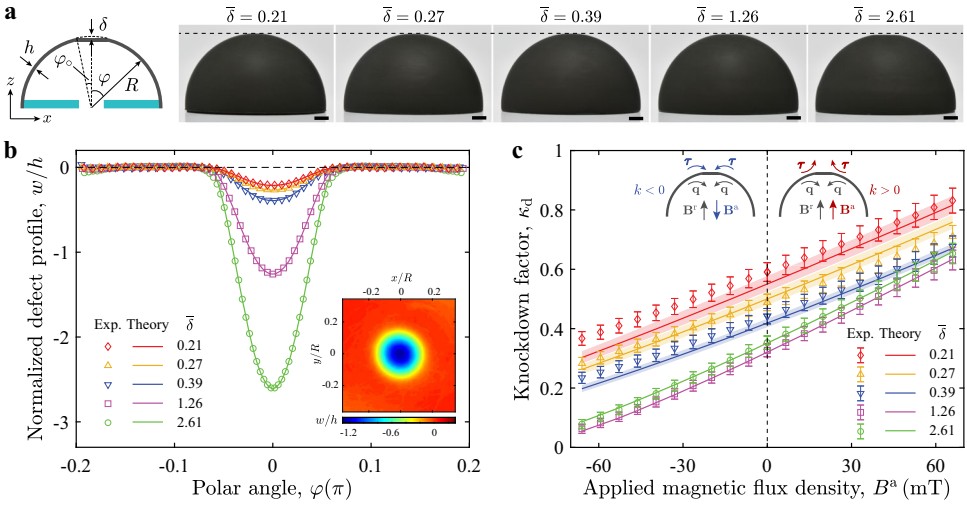

**Fig. 2 Tuning the knockdown factor of magnetic shells with different defect geometries. a** Photographs of a series of magnetic shells with different defect amplitudes, $\overline{\delta} = \{0.21, 0.27, 0.39, 1.26, 2.61\}$, but the same defect width, $\overline{\varphi}_{\circ} = 3.2$. Scale bars are 5 mm. The schematic illustrates the geometric parameters of the shell. **b** Axisymmetric 2D defect profiles normalized by shell thickness, $w/h$, for the shells presented in **a**. Symbols are the latitude-wise average of the corresponding 3D profiles measured through profilometry, and lines are profiles described by $w/h = -\overline{\delta}(1 - \varphi^2/\varphi_{\circ}^2)^2$ $(0 \le \varphi \le \varphi_{\circ})$ (see Supplementary Note 1 for details). Inset: measured 3D defect profile of the shell with $\overline{\delta} = 1.26$ and $\overline{\varphi}_{\circ} = 3.2$. **c** Knockdown factor, $\kappa_{\mathrm{d}}$, versus flux density of the applied field, $B^{\mathrm{a}}$, for shells containing the defects shown in **b**. Symbols and lines represent experimental results and theoretical predictions from the axisymmetric shell model, respectively. The error bars in the experimental data correspond to the accuracy of pressure measurements and the standard deviations of the measurements of shell thickness and Young's modulus; the error bands of theoretical predictions correspond to the standard deviations of the measurements of defect amplitude and shell thickness. Insets: schematics of directions of the shell rotation vector, **q**, and the magnetic torque vector, $\boldsymbol{\tau} = -k\mathbf{q}$, near the pole, for the cases of $\mathbf{B}^{\mathrm{a}}$ opposite (left) or parallel (right) to $\mathbf{B}^{\mathrm{r}}$.

defect amplitudes are varied as $\overline{\delta} = \{0.21, 0.27, 0.39, 1.26, 2.61\}$ during manufacturing (Methods section and Supplementary Note 1). The axisymmetric defect profiles, $w$, are presented in Fig. 2b, defined as the radial deviation of the measured shell profile from a perfect hemisphere. Figure 2c presents the corresponding knockdown factor measurements, $\kappa_{\mathrm{d}} = \overline{P}_{\max}$, as a function of the external flux density, $B^{\mathrm{a}}$. Naturally, due to imperfection sensitivity, the shells exhibit distinct knockdown factors for different values of $\overline{\delta}$. However, in the presence of the magnetic field, we consistently observe an increasing or decreasing knockdown factor over the explored range of $\overline{\delta}$. Within the range of $B^{\mathrm{a}}$ accessible in our experiments, $\kappa_{\mathrm{d}}$ can be changed up to $\approx \pm 30\%$, with respect to the no-field case. These experimental results demonstrate that the knockdown factor of a shell can be dynamically tuned, as an extrinsic quantity, by adjusting the polarity and strength of the applied magnetic field, via a robust mechanism that is insensitive to imperfections.

**Theory of axisymmetric hard-magnetic shells.** To rationalize the experimental results presented above, we develop a theoretical model that predicts the response of hard-magnetic axisymmetric thin shells under a combination of mechanical and magnetic loading. We consider the Helmholtz free energy of ideal hard-magnetic soft materials[49,50], comprising an elastic energy term (related to material deformation) and a magnetic energy term (describing the work to align the residual magnetic vector along the external magnetic field). The shell model is developed by reducing this three-dimensional energy to the 1D profile curve of the middle surface of the shell (detailed derivation provided in Supplementary Note 3). The dimensional reduction of the Kirchhoff-Saint Venant elastic energy[55], valid for small strains and large displacements, with the potential of live pressure was reported recently[56]. In the present study, we focus on the magnetic energy term, which, for 3D scale-free materials, is written as[49,50,57] $\mathcal{U}_{\mathrm{m}} = -\mu_0^{-1} \int \mathbf{F}\mathbf{B}^{\mathrm{r}} \cdot \mathbf{B}^{\mathrm{a}} \, \mathrm{d}V$, where $\mathbf{F}$ is the deformation

gradient[55]. We note that, in the magnetic energy of hard-magnetic materials, $\mathbf{B}^{\mathrm{r}}$ is the remanent magnetization retained in the material (through the saturated hard-ferromagnetic particles), which is independent of the external field $\mathbf{B}^{\mathrm{a}}$. This independence between $\mathbf{B}^{\mathrm{r}}$ and $\mathbf{B}^{\mathrm{a}}$ in hard-magnetic materials contrasts with superparamagnetic and soft-ferromagnetic systems[35–37,40,46,47], where the material magnetization is induced by the field applied for actuation, thereby relating to $\mathbf{B}^{\mathrm{a}}$. In this model, the magnetic interaction between particles embedded in the MRE is not taken into account due to its negligible influence on the results, which has been validated through experiments (evidence provided in Supplementary Note 3). We describe the axisymmetric shell profile by the coordinates $(\mathring{\rho}, \mathring{z})$, $\mathring{\rho}$ being the radial coordinate in the $x$-$y$ plane perpendicular to the axis of axisymmetry ($z$). Accented quantities (e.g., $\mathring{a}$) refer to the undeformed configuration of the shell. The reduced 1D magnetic energy, normalized by $\pi EhR^2/(4(1-\nu^2))$, can be derived as (Supplementary Note 3)

$$\overline{\mathcal{U}}_{\mathrm{m}} = -\frac{8(1-\nu^2)}{R^2} \frac{B^{\mathrm{r}}B^{\mathrm{a}}}{\mu_0 E} \int_0^{\pi/2} \mathcal{F}(\mathring{\rho}_{,\varphi}, \mathring{z}_{,\varphi}, \rho_{,\varphi}, z_{,\varphi}) \, \mathring{a} \, \mathrm{d}\varphi , \quad (3)$$

where $\mathring{a}$ is the area measure, and $\mathcal{F}$ is a dimensionless function that depends on the initial and deformed configurations of the shell. From Eq. (3), we identify a magneto-elastic parameter

$$\lambda_{\mathrm{m}} = \frac{B^{\mathrm{r}}B^{\mathrm{a}}}{\mu_0 E} , \quad (4)$$

which represents the intrinsic magneto-elastic coupling of the system. Equilibrium equations can be generated by minimizing the total energy for all the possible displacements of the shell, which were solved via the Newton-Raphson method (see Methods section and Supplementary Note 3)[56].

It is important to highlight that, in our simulations of the 1D model presented above, we use the geometric and physical parameters of the system measured in experiments (see Methods section), without any fitting parameters. The defect

profile in the region $0 \leq \varphi \leq \varphi_\circ$ is described analytically by $w/h = -\overline{\delta}(1 - \varphi^2/\varphi_\circ^2)^2$ (Fig. 2b), derived based on a simple plate model (Supplementary Note 1). Excellent agreement is found between theory and experiments (Fig. 2c). The variation of the knockdown factor for shells with different defect geometries under the magnetic field is accurately predicted by our shell model, which is, therefore, able to describe the intricate coupling between elasticity, magnetism, and the nonlinear mechanics of thin shells.

**Physical interpretation of the reduced magnetic energy.** Even though our theoretical model can predict the buckling strength of shells, the reduced magnetic energy is highly nonlinear, and the mechanism underlying the change of knockdown factors still needs to be clarified. Therefore, we proceed to expand the integrand of the reduced magnetic energy in Eq. (3), $\mathcal{F}(\overset{\circ}{\rho}_{,\varphi}, \overset{\circ}{z}_{,\varphi}, \rho_{,\varphi}, z_{,\varphi})$, up to second order in the displacement field (Supplementary Note 3). By examining the role of each term in the buckling instability, we conclude that only the following second-order term dominates the buckling of the shell,

$$\overline{\mathcal{U}}_m^{(2)A} = -\frac{8(1-\nu^2)}{R^2} \int_0^{\pi/2} \frac{1}{2}\boldsymbol{\tau} \cdot \mathbf{q} \, \overset{\circ}{a} \, d\varphi , \qquad (5)$$

where $\mathbf{q}$ is the rotation vector of a material fiber of the shell[2,58], and $\boldsymbol{\tau} = -k(\varphi)\mathbf{q}$ can be interpreted as a distributed (dimensionless) torque applied by the external magnetic field. This torque is a linear function of the shell rotation with a deformation-independent pre-factor $k = \lambda_m \overset{\circ}{\chi}(\varphi)$, which we can interpret as the (dimensionless) stiffness of distributed rotational springs, where $\overset{\circ}{\chi}(\varphi) = \overset{\circ}{\rho}_{,\varphi}^2/(\overset{\circ}{\rho}_{,\varphi}^2 + \overset{\circ}{z}_{,\varphi}^2)$.

Under pressure loading, material fibers tend to rotate prior to buckling, thereby increasing the magnetic torque (Supplementary Note 3). Whether the torque reacts to restore the undeformed orientation of the material fibers or to rotate them further away from the initial orientation, depends on the sign of the equivalent stiffness $k$ (insets in Fig. 2c). When $\mathbf{B}^a$ and $\mathbf{B}^r$ are in the same direction ($\mathbf{B}^a \cdot \mathbf{B}^r > 0$), the stiffness $k$ is positive, thus ensuring that $\boldsymbol{\tau}$ is opposite to $\mathbf{q}$, thereby counteracting buckling. As a result, we observe the strengthening of shells with increasing critical loads (cases with $B^a > 0$ in Fig. 2c). Conversely, when $\mathbf{B}^a$ and $\mathbf{B}^r$ are in opposite directions ($\mathbf{B}^a \cdot \mathbf{B}^r < 0$), the negative stiffness ($k < 0$) leads to a torque $\boldsymbol{\tau}$ that acts to increase the rotation $\mathbf{q}$. Indeed, in this

regime, buckling occurs at lower pressure levels (cases with $B^a < 0$ in Fig. 2c).

**Scaling analysis of the change of knockdown factor.** With a physical interpretation of the magnetic energy for shells at hand, we now employ scaling arguments to more clearly rationalize how the knockdown factor is modified by the magnetic field for different radius-to-thickness ratios. Since the magnetic energy interacts with the live pressure potential to alter the knockdown factor, we balance the two energy terms. The dimensionless magnetic energy can be shown to scale as $\overline{\mathcal{U}}_m \sim \lambda_m h/R$, while the change of knockdown factor with respect to the non-magnetic case, $\Delta\kappa_d = \kappa_d - \kappa_d|_{B^a=0}$, scales as $\Delta\kappa_d \sim (\Delta p/E)(R/h)^2$ (Supplementary Note 3). By balancing the scalings for the potential of $\Delta p$, that is $\overline{\mathcal{U}}_{\Delta p} = \Delta p/E$, and the magnetic energy $\overline{\mathcal{U}}_m$, we find $\Delta p/E \sim \lambda_m h/R$, translating into

$$\Delta\kappa_d \sim \frac{\Delta p}{E}\left(\frac{R}{h}\right)^2 \sim \lambda_m \frac{R}{h} . \qquad (6)$$

From Eq. (6), we define the magneto-elastic buckling number,

$$\Lambda_m = \lambda_m \frac{R}{h}, \qquad (7)$$

which governs the knockdown factor under combined pressure and magnetic loading of magnetic shells. The scaling $\Delta\kappa_d \sim \Lambda_m$ provides a scale-invariant description of how the magnetic field modifies the buckling pressure for shells with different radius-to-thickness ratios.

**Robustness of the mechanism to geometric imperfections.** We set out to investigate the role of imperfections in the buckling of magnetic shells, which we now address with a more systematic parametric exploration. We fabricate shells over a wide range of the defect amplitude ($0.1 \leq \overline{\delta} \leq 3.4$) and measure the corresponding knockdown factors, $\kappa_d$, from buckling tests. In parallel, we run 1D simulations for the same material and geometric properties. Figure 3a illustrates the relationship between $\kappa_d$ and $\overline{\delta}$ at different levels of external flux density, included in the magneto-elastic buckling number, $\Lambda_m$, of Eq. (7). The signature of imperfection sensitivity is still observed in the presence of the magnetic field: $\kappa_d$ decreases dramatically when the defect amplitude increases in the regime of relatively small defects ($0 < \overline{\delta} < 1$). Surprisingly, the results of the change in knockdown factor under the applied magnetic field ($\Delta\kappa_d$) presented in Fig. 3b are

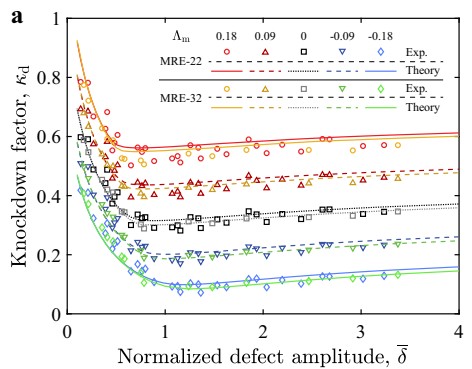
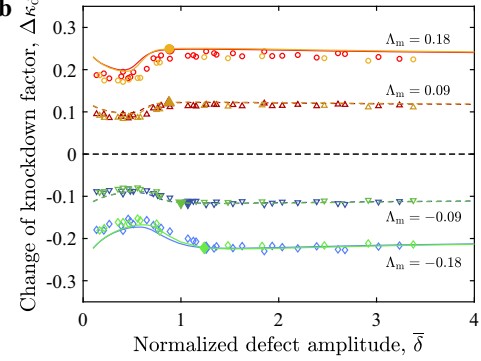

**Fig. 3 Imperfection sensitivity of the pressure buckling of magnetic shells. a** Knockdown factor, $\kappa_d$, and, **b** its change under the magnetic field, $\Delta\kappa_d = \kappa_d - \kappa_d|_{B^a=0}$ are plotted versus normalized defect amplitude $\overline{\delta}$. The tested shell specimens are made of MRE-22 or MRE-32 ($E = 1.15$ MPa, $R/h = 79.1$ for MRE-22 and $E = 1.69$ MPa, $R/h = 91.3$ for MRE-32; see Methods section). The magneto-elastic buckling number $\Lambda_m = \{0.18, 0.09, 0, -0.09, -0.18\}$ is varied by adjusting the external flux density $B^a = \{-53, -26, 0, 26, 53\}$ mT for the MRE-22 shells and $B^a = \{-66, -33, 0, 33, 66\}$ mT for the MRE-32 shells. Experimental results are represented by open symbols, and lines are predictions by the axisymmetric shell model. The closed symbols on each line in **b** represent the onset of the plateau of $\Delta\kappa_d$, determined as the first point when the derivative of $\Delta\kappa_d$ with respect to $\overline{\delta}$ is <1%.

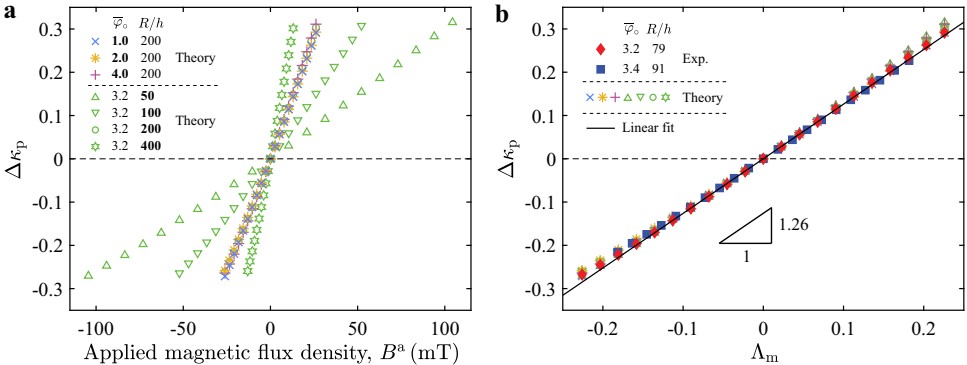

**Fig. 4 Plateau of the change of knockdown factor. a** Plateau change of knockdown factor, $\Delta\kappa_p$, as a function of external magnetic flux density, $B^a$, is presented for shells with different defect widths, $\overline{\varphi}_\circ = \{1.0, 2.0, 3.2, 4.0\}$, and in different radius-to-thickness ratios, $R/h = \{50, 100, 200, 400\}$ (as detailed in the legend). Results are obtained by the axisymmetric shell model. **b** Collapse of the data by plotting $\Delta\kappa_p$ versus the magneto-elastic buckling number $\Lambda_m$. Experimental and theoretical results are juxtaposed, showing excellent agreement. A linear fit on the data indicates a slope of $1.26 \pm 0.01$.

significantly less sensitive to defects; $\Delta\kappa_d$ becomes approximately constant for $\overline{\delta} > 1$. This finding suggests that the magnetic interaction between the shell and the external field is nearly unaffected by the intrinsic imperfection sensitivity, ensuring a robust tuning of the knockdown factor.

Moreover, the results in Fig. 3a, b include data for two sets of shells with different combinations of material and geometric properties ($E = 1.15$ MPa, $R/h = 79.1$ for MRE-22 shells and $E = 1.69$ MPa, $R/h = 91.3$ for MRE-32 shells; see Methods section). Still, the $\kappa_d$ and $\Delta\kappa_d$ data collapse for these two sets, given that the value of $\Lambda_m$ is the same for both. This collapse supports $\Lambda_m$ as the governing parameter, combining the mechanical, magnetic, and geometric properties of the system. Throughout the analysis, the theory is in excellent agreement with experiments.

A robust way to quantify the effect of the magnetic field on the change in knockdown factor is to focus on the plateau in Fig. 3b, $\Delta\kappa_p$, defined as the average of $\Delta\kappa_d$ over the extent of the plateau-like region. Figure 4a (open symbols) shows predictions from our shell model for $\Delta\kappa_p$ versus the external flux density, $B^a$, for shells with different radius-to-thickness ratios and defect widths. While the radius-to-thickness ratio strongly influences the change of knockdown factor, the defect width has little effect on it (Supplementary Note 4). In Fig. 4b, we plot the raw data of Fig. 4a as a function of the magneto-elastic buckling number $\Lambda_m$, finding that the different results for shells with different geometries fall onto a master curve. This collapse demonstrates that the plateau change of knockdown factor is governed by the single parameter $\Lambda_m$, independent of the geometry of the defect. The superposition of experimental results on the theoretical predictions further validates this collapse. Moreover, consistently with the scaling in Eq. (6), a linear relationship is found between $\Delta\kappa_p$ and $\Lambda_m$, with a slope of $1.26 \pm 0.01$ obtained via linear fitting. This master curve serves as a concrete design guideline for magnetic shells, summarizing the effect of the magnetic field on tuning the buckling strength of pressurized spherical shells with different material and geometric properties.

## Discussion

Thus far, we have demonstrated the possibility of tuning the knockdown factor of axisymmetric shells via magnetic actuation. This change of knockdown factor is dictated by the magnetic torque generated due to the shell rotation, which breaks the alignment between the magnetization ($B^r$) and the applied field ($B^a$). Consequently, this mechanism is expected to be preserved for asymmetric loading conditions, provided $B^r$ and $B^a$ are aligned in the initial configuration. To attest this statement, we

first magnetize our shell specimens asymmetrically, at an angle $\varphi^r$ with respect to the shell's axis of symmetry (see schematic diagram in Fig. 5a). The defect is still maintained at the pole. Then, during the buckling test, a magnetic field at the same angle ($\varphi^a = \varphi^r$) is applied, either parallel ($B^r \cdot B^a > 0$) or antiparallel ($B^r \cdot B^a < 0$) to the initial shell magnetization. In Fig. 5a, we plot the measured change of knockdown factor, $\Delta\kappa_d$, versus $\varphi^r$ and $\varphi^a$, at $\Lambda_m = \pm 0.22$. We consistently observe finite values of $\Delta\kappa_d$ for both the axisymmetric and asymmetric cases, even when $B^r$ and $B^a$ are perpendicular to the axis of the defect. In the latter, the most unfavorable case, the range of $\approx \pm 0.1$ in the tuning of $\kappa_d$ is still significant. These results indicate that, even when the defect information might be unknown in an application setting, the tunable mechanism that we have uncovered can always be realized by aligning the applied field with the magnetization. Still, at the same level of $\Lambda_m$, the capacity of tuning decreases with increasing $\varphi^r$ and $\varphi^a$, and the axisymmetric configuration is the most efficient at maximizing $\Delta\kappa_d$.

To further probe the robustness of the proposed mechanism, we consider asymmetric cases, where $B^r$ and $B^a$ are misaligned. We focus on shell specimens with a fixed magnetization set at $\varphi^r = 0°$, while the direction of the applied field is varied by increasing $\varphi^a$ from $0°$ (axisymmetric, $B^a \| B^r$) to $90°$ ($B^a \perp B^r$). The corresponding change of knockdown factor is plotted in Fig. 5b, which (the absolute value) decreases from the maximum at $\varphi^a = 0°$ to $\Delta\kappa_d = 0$ at $\varphi^a = 90°$. Thus, in settings where the alignment between the shell magnetization and the external field cannot be ensured, tuning the knockdown factor is always possible (unless $B^a \perp B^r$, which leads to $\Delta\kappa_d = 0$). This relaxation in loading conditions broadens the applicability of the mechanism in real applications involving complex magnetization profiles and magnetic fields. In parallel to the experiments, we perform full 3D finite element simulations (see Methods section for details) using a user-defined element proposed in Zhao et al.[50] for hard-magnetic soft materials. Good agreement between simulations and experiments is found in Fig. 5a, b, which further corroborates our findings. To establish quantitative understanding on the magneto-elastic interaction in the asymmetric cases, a general magnetic shell model would be required, which is beyond the scope of the present study. We also note that, since the generation of magnetic torque is not constrained to a specific defect geometry, we anticipate our mechanism would also work for imperfections with other geometric profiles or arrangements. The localized nature of buckling in spherical shells[19,20] is also underlined in our system by the localized distribution of shell rotation and the associated magnetic torque in the vicinity of the defect (see Supplementary Fig. 6 in Supplementary Note 3). This

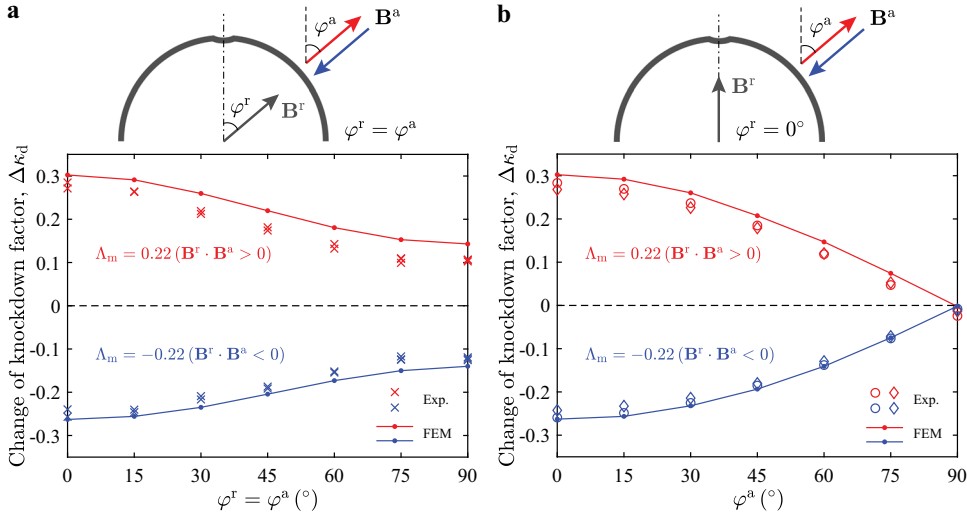

**Fig. 5 Tunability under asymmetric loading conditions. a** Change of knockdown factor, $\Delta\kappa_d$, at $\Lambda_m = 0.22$ ($\mathbf{B}^r \cdot \mathbf{B}^a > 0$) and $\Lambda_m = -0.22$ ($\mathbf{B}^r \cdot \mathbf{B}^a < 0$), versus angle of the shell magnetization, $\varphi^r$, and of the applied field, $\varphi^a$, which are consistently aligned in the initial configuration ($\varphi^a = \varphi^r$). In the experiments, 16 shell specimens containing a defect at the pole with identical geometry are fabricated (with average defect amplitude $\bar{\delta} = 1.64 \pm 0.04$ and defect width $\bar{\varphi}_\circ = 3.2$). The shells are magnetized at an angle $\varphi^r$ with respect to the axis of symmetry (2 or 3 specimens at each angle, as indicated by the number of data points). Each specimen (magnetized at $\varphi^r$) is tested with the magnetic field oriented at $\varphi^a = \varphi^r$. **b,** Change of knockdown factor, $\Delta\kappa_d$, at $\Lambda_m = 0.22$ ($\mathbf{B}^r \cdot \mathbf{B}^a > 0$) and $\Lambda_m = -0.22$ ($\mathbf{B}^r \cdot \mathbf{B}^a < 0$), as a function of $\varphi^a$, with fixed $\varphi^r = 0$. The two shell specimens used for the data in **b** are the same as those from **a** that are magnetized at $\varphi^r = 0$.

localization ensures that the tunability would not be affected by the shell opening angle and boundary conditions (except for extremely shallow shells).

In closing, we have shown that the buckling strength of shells can be dynamically tuned by exploiting the interplay between mechanics and magnetism. The proposed mechanism represents a robust way to gain control on a property of thin shells that has long been regarded as intrinsic, the knockdown factor. By performing precision experiments on hard-magnetic elastomeric shells and developing a theoretical model, we unveiled the essential feature at the base of the mechanism, a distributed torque induced by the magnetic field due to the shell rotation. Moreover, we showed that a dimensionless quantity, the magneto-elastic buckling number, emerges as the key governing parameter, summarizing the geometric, mechanical, and magnetic properties of the system. We envision that our tunable mechanism can be used to gain control on the structural life of a shell, by applying a magnetic field as critical conditions approach. More generally, we believe that the approach of coupling mechanical deformation and magnetic actuation, already successfully applied for beams[33,48,51] and films[47], and employed here to tune the knockdown factor of shells, can be extended to modify the instabilities of other structures such as rods, plates, and non-axisymmetric shells[45,46], for which research efforts are currently ongoing. We hope that the principle that we have uncovered will open an axis of tunability in the design of mechanical systems where shell buckling is harnessed as a functional mechanism[59–61], to offer devices with tunable mechanical properties or behavior.

## Methods
**Preparation of the MRE material**. Our experimental samples were fabricated using a MRE material composed of a mixture of hard-magnetic NdPrFeB particles (average size of 5 μm, MQFP-15-7-20065-089, Magnequench) and Vinylpolysiloxane (VPS, Elite Double, Zhermack), a silicone-based polymer. MREs made of VPS Elite Double 22 or VPS Elite Double 32 are referenced throughout the text as MRE-22 or MRE-32, respectively. We prepared the MRE with the following steps. First, the initially non-magnetized NdPrFeB particles were mixed with the liquid VPS base (1:1 mass ratio) using a centrifugal mixer (ARE-250, Thinky Corporation); for 40 s at 2000 r.p.m. (mixing mode), and another 20 s at 2200 r.p.m. (defoaming mode). Secondly, the solution was degassed in a vacuum chamber

(absolute pressure below 8 mbar), and then cooled down to room temperature (22 ± 0.4 °C) to avoid any changes of viscosity due to thermal disturbances from the previous steps. Thirdly, VPS catalyst was added to the mixture, with a ratio of 1:1 in weight to the VPS base. After another mixing step for 20 s at 2000 r.p.m. (mixing mode) followed by 10 s at 2200 r.p.m. (defoaming mode), the final mixture was ready for sample fabrication. The final mass concentrations of NdPrFeB particles and VPS polymer were 33.3 wt% and 66.7 wt%, respectively. Pouring of the liquid MRE mixture onto the mold during shell fabrication (see below) was done after a set waiting time (100 s for MRE-22 and 20 s for MRE-32), so as to increase the viscosity to the desired value. After the above preparation steps, the curing of the polymer mixture occurred in ~20 min at room temperature.

**Physical properties of the MRE**. The densities of MRE-22 and MRE-32 were measured using a pycnometer to be $\rho_{MRE-22} = 1.61 \pm 0.05$ g/cm$^3$ and $\rho_{MRE-32} = 1.63 \pm 0.03$ g/cm$^3$, respectively. The density of the NdPrFeB particles was 7.61 g/cm$^3$ (provided by the supplier). Under the concentrations of the prepared mixture, the volume fraction of NdPrFeB particles was calculated to be 7.05% for MRE-22 and 7.14% for MRE-32. When saturated, the effective residual flux density of our MRE, $B^r$, was assumed to be the volume-average of the total residual flux density of individual NdPrFeB particles (0.90 T, as reported by the supplier). Given the chosen volume fraction, $B^r = 63.2$ mT for MRE-22 and $B^r = 64.0$ mT for MRE-32. The Young's moduli of MRE-22 and MRE-32 were measured to be $E = 1.15 \pm 0.04$ MPa and $E = 1.69 \pm 0.06$ MPa, respectively, using a combination of cantilever and tensile tests.

**Fabrication of the magneto-active shell specimens**. Our shells were fabricated by coating the underside of a flexible negative spherical mold (radius 25.4 mm) with liquid MRE; see the schematic in Supplementary Fig. 1 and Supplementary Note 1 for details. Whereas our technique was inspired by previous work[17,18], our molds also contained a soft spot (thin circular region) at the pole to produce a precisely engineered defect. With the negative spherical surface of the mold facing down, the gravity-driven viscous flow of the polymer yielded a thin layer of MRE on the mold. Before the MRE fully cured (≈13 min after pouring), the mold was depressurized from within using a syringe pump. As a result, the soft spot of the mold deformed to produce an axisymmetric geometric imperfection at the pole of the shell. This defect became frozen in the shell upon curing. The amplitude of the defect was systematically varied by adjusting the pressure imposed on the mold during curing (Supplementary Fig. 2). The width of the defect can be varied by changing the pillar used in mold manufacture with a different radius (Supplementary Note 1). The amplitude and width of the defect were set independently. To make the shell magnetically active, we saturated the NdPrFeB particles, already in the solidified MRE, using an applied uniaxial magnetic field (4.4T, perpendicular to the equatorial plane) generated by an impulse magnetizer (IM-K-010020-A, Magnet-Physik Dr. Steingroever GmbH).

**Geometry of the shells**. The radius of the shells ($R = 25.4$ mm) was set by the mold used during fabrication. Their thickness was characterized using a microscope (VHX-5000, Keyence Corporation) after cutting off narrow strips ($\approx 2 \times 6$ mm$^2$) near the pole. The average measured thickness was $h = 321.2 \pm 5.1$ μm and $h = 278.2 \pm 2.8$ μm for the MRE-22 and the MRE-32 shells, respectively. Hence, the corresponding radius-to-thickness ratios were $R/h = 79.1$ and $R/h = 91.3$. The 3D profile of the outer surface of each fabricated imperfect shell was characterized using an optical profilometer (VR-3200, Keyence Corporation). The geometry of the defect was computed as the radial distance between the measured profile and the corresponding best-fit spherical surface. Specifically, the amplitude, $\delta$, and half angular width, $\varphi_\circ$, of the defect were determined by fitting the analytical description, $w/h = -\overline{\delta}\left(1 - \varphi^2/\varphi_\circ^2\right)^2$ ($0 \leq \varphi \leq \varphi_\circ$), to the experimental measurements (additional details provided in Supplementary Note 1). The ranges of geometric parameters for our specimens were as follows: $0.14 \leq \overline{\delta} \leq 3.2$ and $\varphi_\circ = 11.7 \pm 0.1°$ ($\overline{\varphi}_\circ = 3.2$) for the MRE-22 shells; and $0.17 \leq \overline{\delta} \leq 3.4$ and $\varphi_\circ = 11.6 \pm 0.1°$ ($\overline{\varphi}_\circ = 3.4$) for the MRE-32 shells.

**Generation of the magnetic field**. A uniaxial uniform magnetic field was generated in the central region of two identical customized multi-turn circular coils (square cross-section, inner diameter 72 mm, outer diameter 114 mm, and height 21 mm), configured in the Helmholtz configuration. The coils were set concentrically, with a center-to-center axial distance of 46.5 mm, and connected in series, powered by a DC power supply providing a maximum current/power of 25 A/1.5 kW (EA-PSI 9200-25 T, EA-Elektro-Automatik GmbH). The flux density of the magnetic field was varied in the range $-66$ mT $\leq B^a \leq 66$ mT by adjusting the current output of the power supply, from $-10$ A to 10 A. The flux density was measured using a Teslameter (FH 55, Magnet-Physik Dr. Steingroever GmbH), along both the axial and radial directions. In parallel to the experimental measurements, we simulated the generated field using the Magnetic Fields interface embedded in the commercial package COMSOL Multiphysics (v. 5.2, COMSOL Inc.), based on Ampère's Law (see Supplementary Note 2 for details). Excellent agreement was found between experiments and simulations (Supplementary Fig. 3).

**Buckling tests**. To quantify the critical buckling conditions of our magnetic shells, we positioned each shell in between the Helmholtz coils under a steady magnetic field, and then depressurized it under prescribed volume conditions by a pneumatic-loading system (see Supplementary Note 2 for details). The applied pressure was monitored by a pressure sensor (785-HSCDRRN002NDAA5, Honeywell International Inc.). The level of depressurization was increased, under a steady magnetic field, until the shell buckled. The critical buckling pressure was defined as the maximum value of the loading curve (pressure vs. imposed volume change).

**Theory and numerics**. The reduced 1D energy $\overline{\mathcal{U}}$, representing the sum of the elastic and magnetic energies of the system, together with the potential of the live pressure, was obtained by means of dimensional reduction (see Supplementary Note 3 for details). Equilibrium equations were generated by minimizing the total energy for all possible displacements of the shell, and solved via the Newton-Raphson method using the commercial package COMSOL Multiphysics (v. 5.2, COMSOL Inc.). Details of the implementation of our numerical procedure are provided in Pezzulla and Reis[56].

**Finite element modeling**. Finite element simulations were performed using the software package Abaqus/Standard, with no fitting parameters; all the geometric, mechanical, and magnetic parameters required in the simulations were measured independently from the experiments. Geometric nonlinearities were taken into account. Due to the symmetry of the system, only half of the hemispherical shell (i.e., one quarter of a full spherical shell) was modeled, applying symmetric boundary conditions on the $x-z$ plane. The geometric defect described by the theoretical profile (see Supplementary Note 1 for details) was introduced into the shell. The shell was discretized by the user-defined 3D 8-node isoparametric solid elements proposed in Zhao et al.[50], for modeling hard-magnetic materials. The mechanical response of the material was described by the incompressible neo-Hookean model with a bulk modulus of $100E$ to realize the incompressibility. A convergence study was performed to select the level of discretization of the mesh with eight elements in the thickness direction, 200 elements along the half equator, and 200 elements along the half meridians. During loading, the shell was clamped at the equator with the magnetic field applied in a first step. In a second step, the shell was pressurized by imposing a uniform live pressure on a duplicate dummy mesh of C3D8RH solid elements, congruent (sharing the same nodes) to the mesh of the user elements. The elastic modulus of this dummy material was set to $10^{-20}$ Pa, ensuring nearly vanishing element stiffness and elastic energy of this dummy shell with respect to the physical shell. This second step was conducted using the Riks method to capture the critical load of subcritical bifurcation.

## Data availability

Data supporting the findings of this study are available within the paper and its Supplementary Information files. All other relevant data are available from the corresponding author upon reasonable request. Source data are provided with this paper as a Source Data file. Source data are provided with this paper.

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

## Acknowledgements

We thank Selman Sakar and Lucio Pancaldi-Giubbini for providing the coils. We acknowledge Benjamin Rahm for help with preliminary experiments. A.A. is grateful to the support from the Federal Commission for Scholarships for Foreign Students (FCS) through Swiss Government Excellence Scholarship (Grant No. 2019.0619).

## Author contributions

D.Y. and P.M.R. conceived the project. D.Y., L.C., and A.A. performed experiments and analyzed data. M.P. developed the theoretical model and performed the scaling analysis. M.P and D.Y. performed simulations and analyzed data. P.M.R. supervised the research. D.Y., M.P., A.A., and P.M.R. wrote the paper.

## Competing interests

The authors declare no competing interests.
