## [Peer Review File · Nature Communications]

Reviewers' Comments:

Reviewer #1:

Remarks to the Author:

The first question I have after reading this nicely written paper is why one wants to use a magnetic field to tune the buckling? In the paper, the authors chose a symmetric magnetization profile and a geometric defect at the top, which can be simplified using a one-dimensional model. This simple setup allows the experiments to be highly repeatable with an error below 10%. This observation may seem a little contradictory to the introduction, that the buckling is sensitive to imperfection and the knockdown factor is difficult to predict. It also raises an important question as to why one needs to use an external magnetic field to tune the knockdown factor instead of the thickness of the shell or the elastic modulus, if the imperfections can be measured precisely. Fundamentally, using a magnetic field is changing the "effective modulus" of the soft materials, one can also tune the effective modulus by using polymers that respond to thermal, acoustic, light, and other stimuli.

The second question is how much the conclusion can be generalized. In most engineering applications, the shells are not a half sphere and the geometric imperfection is not a well-defined thickness at the top of the shell. Can we expect the tunability of the magnetic field valid? How far we can stretch the conclusion of this paper? I am not asking for more experiments, maybe the authors can provide more insights and interpretation of their results, considering the broad readership of Nature Communications.

Another thing I note is that the magnetization profile is symmetric in the study. What if the shell is magnetized along the x-direction as in Figure 1c (also the Helmholtz coil is rotated 90 degrees to apply horizontal magnetic field)? For magnetic elastomer actuators, the magnetization profiles are often complex and will deform under an external magnetic field. The authors are encouraged to provide more insights into what will happen for the shells under deformation.

Reviewer #2:

Remarks to the Author:

The stated purpose of this manuscript is

Here, we propose a robust mechanism to dynamically tune the buckling strength of shells (i.e., their knock-down factor), by coupling elastic deformation and magnetic actuation, to provide active control and liberate the predestination of imperfections.

Toward that, the authors have shown that the buckling strength of shells can be dynamically tuned by exploiting the coupling between mechanics and magnetism. The authors used a combination of bench scale precision experiments and theoretical modeling on hard-magnetic elastomeric shells. They show that the tuning mechanism is a distributed torque induced by the magnetic field throughout the shell. And they also show that a dimensionless quantity, the magneto-elastic buckling number, emerges as the key governing parameter, summarizing the geometrical, mechanical, and magnetic properties of the system. The results are novel, and showcase how buckling may be tuned using applied magnetic fields. Overall the manuscript is in an excellent condition and only needs to consider the following minor comments:

1) In S1.2, the authors state:

We used a simple model to provide an analytical description of the geometry of the defect. Note that, during pressurization of the mold, the bulk region outside of the soft spot exhibits negligible deformation compared to the soft spot. Moreover, the spherical cap of this soft spot is sufficiently

small (with respect to the radius of curvature of the shell), such that it can be regarded as a (nearly) flat plate. Hence, we isolate this thin region and model it as a boundary-clamped circular flat plate loaded by a uniform pressure.

Is there any possible quantification to help support the (reasonable) assumption for the clamped boundary condition?

2) In S2.2, the second paragraph under 'Protocol of the buckling test' states

Noting that the flow rate of 0.4mL/min for the pressurization was set constant for all experiments, these experimental could run for up to 70s, which was far too long for the coils to be switched on.

but the word experimental is most probably intended to be experiments.

3) In S2.2, the second paragraph under 'Protocol of the buckling test.' Can some description be added to the heating of the pressurized air as a motivation for not continuously applying the magnetic field. I do not know much about these coils, and why they would heat up the air in the pneumatic system.

Reviewer #3:

Remarks to the Author:

This paper presents a new means of controlling the buckling threshold of shells subjected to an external pressure. The idea is that shells with tunable mechanical response can be made by adding hard magnetic particles to the polymer mixture used to form a shell - these particles have an intrinsic magnetic moment after an initial magnetisation step, which is referred to as a residual magnetic field in this paper. When the composite structure is then subject an external pressurization then the response of the shell can, to an extent be tuned by the application of an additional field (B^a). The authors demonstrate this by considering the threshold pressure required for buckling as a function of the various parameters in the problem. In particular, they show that this buckling is imperfection-sensitive (a result that is well known from the literature without a magnetic field), and characterize this by measuring a knock-down factor. They present experimental and theoretical results that are in good agreement, showing that they have developed a good understanding of the underlying phenomenon.

I have some technical questions, which I describe below, but my general sense is that the paper is really more suited to a specialist mechanics journal, rather than a general interest one like Nature Communications. I think the idea is a really neat one, but, for the moment, the authors did not convince me that they have a 'killer app' that could be developed with this idea - indeed, the paper finishes a little weakly with the authors supposing that "magnetic structures with predictable and tunable buckling behavior will enrich the design of advanced materials with new functionalities" but do not give more details. This asks follow-up questions: what new functionalities would this enable, and how can this be done when the maximum relative change in knock-down factor is about 50%?

Previous work:

There is a large body of work on two problems that are closely related to that considered here: buckling of elastic objects driven by magnetic fields (magneto-elastic buckling) and the magnetism induced resistance to buckling under other external fields (but no applied magnetic field). At some level, the current paper seeks to combine these two points of view. However, as far as I could tell, this paper misses the main literature in each of these areas. I think it would be appropriate to consider these literatures, and to cite the relevant works appropriately.

(i) There is a long history of work on buckling induced by magnetic fields; I believe the classic work is by Moon and Pao, *Appl Mech* 36, 92 (1968) but there has also been a considerable amount of recent work on this problem ongoing, see Gerbal *Proc. Natl Acad Sci* 112, 7135 (2015) and Cebers, *Phys Rev E* 76, 031504 (2007).

There have also been many studies on the properties of chains of paramagnetic particles, which have an effective tension within the chain because the induced magnetic moments are always aligned with the applied field (see, for example, the work of Roper, Dreyfus et al: *Nature* 436, 862 (2005) and especially section 2.2 of *J. Fluid Mech.* 554, 167 (2006), as well as the work of Cebers, e.g. *Phys Rev E* 70 021404 (2004)).

(ii) The second related problem concerns the properties of chains of ferromagnetic beads. Qualitatively speaking, these have an effective bending stiffness arising from the permanent magnetic dipole, which stiffen such chains from buckling under the influence of gravitational loading (see, for example, work by the groups of Goriely, *Proc R Soc A* 470 20130609 (2014) and Fried, *Proc R Soc A* 473 20160703 (2017)). If I understood these works properly, they seem to have shown that there's an effective Young's modulus E_{eff} proportional to B^2/μ_0 where B is the magnetic field strength of the beads.

Technical questions:

- I am confused about why the magnetic energy of eqn (S4) includes only the interaction between the residual field and that of the applied field? Based on the papers mentioned in (ii) above, there should also be an energetic cost associated with the deformation of the shell because of the interaction of the residual magnetic field with itself (or rather other parts of itself). I originally thought that this could be neglected if $B^r < B^a$, but it seems that this is not the case experimentally. Could the authors discuss this point.

- One of the main results of the paper is the scaling in eqn (6). However, I was not able to understand the scaling analysis that is meant to lead to this result, which is given in section 3 of the SI. For example, in (S51) a complex integral with factors of λ_m/R^4 is comprehensible but is immediately replaced in scaling terms by $\lambda_m h$ after apparently "noticing a linear proportionality to the thickness-to-radius ratio". What exactly does this mean (which terms scale with thickness etc? What happens to the extra factor of R^3 that was in the first portion of (S51)?). I have similar comments/questions about the other scaling laws in this section. Others, I do not understand at all. For example, after (S53) we are told that the volume scales as $R^2 h$. This is not what I would expect for pressure buckling, where the radial scale of the dimple that forms scales as $(Rh)^{1/2}$ and has height h (hence volume $h^2 R$). What did I miss?

- The authors show that the change in knock-down factor induced by changing the parameter λ_m , but is more or less independent of the defect amplitude, δ . This seems very surprising, but I could not really understand from the analysis presented why this should be the case - can the authors offer any physical intuition about this?

- It would be helpful if the authors could relate their work to the other bodies of work mentioned in (i) and (ii) above. For example, the authors' results show that the knockdown parameter is proportional to the magnetic parameter λ_m . It seems that this is almost the same as saying that the effective modulus of the shell is changed from the 'true' material E , to an effective value that supplements the modulus with an additional one proportional to the product of the residual and applied magnetic fields. Is this correct, and can it be understood qualitatively as the effect of a magnetism-induced elasticity, described in (ii) above?

- As far as I could tell, the residual field is always applied in the direction opposing the action of the pressure. I think that flipping the sign of B^r does not change anything, since it is only the sign of the product $B^r B^a$ that matters, but it might be worth noting that this is the case.

Minor comments:

In figure S2 I think the geometrical parameters of the shell (R and h) should be given too, since this is important in determining the critical pressure.

After eqn (S55), it would be helpful to write $\kappa_d \sim (p/E)(R/h)^2$ (brackets added) to be clear about whether the $(R/h)^2$ should be on the denominator or not.

Page 16 of the SI: "compassion" should be "comparison".

Response to the Reviewers' Comments for Manuscript NCOMMS-20-32420-T
"Magneto-active elastic shells with tunable buckling strength"
Dong Yan, Matteo Pezulla, Lilian Cruveiller, Arefeh Abbasi, Pedro M. Reis

Please refer to the revised version of our manuscript for a detailed account of all the changes, corrections and additions; the revised/new text is **marked in blue**. In the following, we address each of the Reviewers' points (**original comments in bold**). The numbering of the comments (e.g., 1.1, 1.2, ..., 2.1, 2.2,...) is added by us to aid in cross-referencing our responses. A list of relevant references, some of which are not included in the manuscript, is provided (e.g., as [R1.1], [R1.2],..., [R2.1], ...) at the end of the response to each Reviewer.

Summary of changes:

- We have performed an additional experimental and computational campaign to explore asymmetric loading conditions. The new results demonstrate that the tunability mechanism is still preserved, pointing to its generality. We have added a new '*Discussion*' section including a new figure 5 in the revised manuscript to report these new results and highlight the broad applicability and generality of the mechanism that we have uncovered in this work.
- We added a new subsection '*Finite element modeling*' in the '*Methods*' section, where we detail the computational approach to model the buckling of hard-magnetic elastomeric shells using the finite element method.
- We have provided more details about the theoretical model we used to describe the defect profiles analytically and about the reason and consequence of the overheating of magnetic coils in the experiments.
- In the introduction of the revised manuscript, we have more thoroughly acknowledged past pioneering work on magneto-elastic systems, while contrasting it to the hard-magnetic materials that we used, which helps highlight the novelty of our work. The corresponding literature has been added to the '*References*' section.
- The recent advances in hard-magnetic materials, which we used in this work, were also emphasized in the modeling part (by specifying the underlying magnetic energy), contrasting to the materials used in past studies.
- We have added several paragraphs in '*Supplementary Note S3.3*' to elaborate on the scaling analysis. The scaling arguments have been clearly clarified in the revised version of the paper.
- When interpreting the tunability, we determined the direction of the torque and the associated change of knockdown factor by the sign of the dot product $\mathbf{B}^r \cdot \mathbf{B}^a$. This general statement is now supported by our new results for asymmetric cases.
- The typos/minor changes pointed out by the reviewers have been corrected/implemented.
- A number of minor changes and corrections of our own accord were implemented through the main text, which we believe help further improve the manuscript.

1. Response to Reviewer #1

We thank the Reviewer for the detailed feedback and suggestions, all of which have been implemented. We believe that these additions to our manuscript have helped further improve its quality. More specifically, the Reviewer's feedback has motivated us to embark on a new extensive experimental and computational campaign to explore non-axisymmetric cases (when the defect is not aligned with the magnetization and/or the applied field). The new results have helped us establish the broader applicability and generality of the magneto-elastic mechanism that we have uncovered. We are grateful and encouraged by the constructive and invaluable review process.

1.1a. The first question I have after reading this nicely written paper is why one wants to use a magnetic field to tune the buckling?

Imperfection sensitivity of shell buckling is a fundamental and longstanding classic problem in structural mechanics, which is also of great importance to the design of a large class of engineering shell-like structures [R1.1] that are inevitably imperfect due to manufacturing or as a cause of continued usage. Despite decades of research, this is still a poorly understood problem. Indeed, toward understanding and predicting the knockdown factor of imperfect shells, a large number of studies have been carried out since the beginning of the last century [R1.2-R1.4]. Recently, thanks to a breakthrough technique in shell fabrication [R1.5], it has been demonstrated, for the first time, that the knockdown factor can be predicted if the imperfections are modeled appropriately [R1.6]. This pioneering study has revived the highly nontrivial field of buckling of doubly curved shells [R1.7-R1.11]. Despite the gained understanding established through these recent studies, it is still not possible to control the knockdown factor, which is dictated by the imperfections that are introduced inevitably during fabrication. As a result, the knockdown factor is regarded as an intrinsic property that cannot be modified post-fabrication. Gaining external predictive control over the buckling strength of shells, which we realize for the first time in the present study, is a fundamental advance in engineering mechanics that we anticipate will find broad usage given the importance of shell stability (and instability) across a variety of applications and length scales.

Our work is fundamental in the sense that we propose a general means to actively control the buckling strength of shells. Indeed, we demonstrate that the critical buckling pressure of thin shells can be manipulated remotely by applying an external magnetic field. This principle brings tunability into mechanical systems where shell buckling might be harnessed as a functional mechanism, thereby opening a new route to enable future design of devices with tunable mechanical properties or behavior; *e.g.*, in soft robots [R1.12,R1.13], metamaterials [R1.14-R1.16], actuators [R1.17,R1.18], and microcapsules [R1.19-R1.21]. For example, using magnetic shells, the buckling instability that has been used to propel spherical swimmers [R1.12] could be controlled remotely by a magnetic field, which would be faster and might be used to adjust the propulsion strength by making the buckling event more/less catastrophic (measured by the pressure drop at the onset of buckling; connected to the results in Fig. 1d of our paper). As a more speculative example, the extreme deformation or breakage of capsules via buckling could also be exploited as a mechanism for drug delivery. If the microcapsule were to be made out of a magnetic material, the onset of buckling could be actuated dynamically, and on demand, by a remote magnetic field and applied in a confined space. In addition, displacement of the capsule could be steered to a specific location by a magnetic field for targeted treatment. We believe that numerous other

innovations may be envisioned to leverage the mechanism that we uncover but we prefer not to over-speculate about this in the manuscript, focusing instead on our fundamental contribution.

Equally important, shell buckling under magnetic actuation is a coupled-field problem in structural mechanics, involving a nontrivial interplay between geometric nonlinearity, mechanical instability, and magnetic actuation. The dimensionless magneto-elastic buckling number identified in this work and the derived scaling laws provide valuable physical insight into this class of problems. The proposed theoretical model establishes the foundation for the rational and predictive design of magnetic shell structures.

We anticipate this paper will be of interest to the broad readership in mechanical, civil and aerospace engineering [R1.1], soft robotics [R1.12,R1.13,R1.17,R1.18], metamaterials [R1.14-R1.16], and biomedicine [R1.19-R1.21], for its fundamental contributions on magnetic thin shells and the potential applications. Furthermore, there is a growing community studying magneto-elastic systems in the past couple of years, but most have been application oriented [R1.22-R1.26]. We believe that it is crucial for these advances to go hand-in-hand with more fundamental developments, as is the case of our manuscript. In fact, we identified Nature Communications as the ideal venue for our manuscript precisely for its broad range in reporting studies along the fundamental-to-applied spectrum (and not just the latter). Our contribution is decidedly towards the former, under the umbrella of engineering science/mechanics.

1.1.b. In the paper, the authors chose a symmetric magnetization profile and a geometric defect at the top, which can be simplified using a one-dimensional model. This simple setup allows the experiments to be highly repeatable with an error below 10%. This observation may seem a little contradictory to the introduction, that the buckling is sensitive to imperfection and the knockdown factor is difficult to predict.

The knockdown factor is difficult to predict in the sense that shell structures in engineering typically contain imperfections with complex geometries and distributions that are hard to be precisely characterized. This inaccurate characterization induces large uncertainties in the prediction of the critical buckling load, due to the imperfection sensitivity. For this reason, before the breakthrough experimental work (Ref. [R1.6]) published in 2016, there were very few experimental studies on shell buckling [R1.4] to quantitatively investigate the imperfection sensitivity, and poor agreement was found regularly between experiments and predictions. To the best of our knowledge, the work in Ref. [R1.6] was the first time that experimental measurements could be accurately predicted by theory and computation. This work benefits from a precision shell fabrication technique in lab settings [R1.5], with which we were able to precisely introduce a defect into a shell and characterize the defect geometry. The most important conclusion of this work is that the knockdown factor can be predicted accurately, provided the defect is modeled thoroughly, the technical details for which were also provided in Ref. [R1.6].

In the current work, we have adapted and augmented the fabrication protocol to fabricate precisely imperfect shells. More specifically, using our cutting-edge high-precision experimental approach, we: (i) fabricated magnetic shells containing an engineered defect, (ii) precisely characterized the geometry of the defect through 3D profilometry (see Fig. 2b in the main article and Supplementary Note S1 for details), and (iii) considered this defect in our theoretical and computational modeling. Thus, we could predict the knockdown factor with high accuracy. More importantly, beyond 'just' prediction (which was unattainable prior to 2016 [R1.6]), we are now

able to tune the knockdown factor, on demand, by a magnetic field, which we believe is a significant advance in the field of shell mechanics that will also be of practical interest in a variety of future applications, as we have detailed in the response to Comment 1.1a.

1.1.c. It also raises an important question as to why one needs to use an external magnetic field to tune the knockdown factor instead of the thickness of the shell or the elastic modulus, if the imperfections can be measured precisely.

In the design stage, one can choose the thickness or the elastic modulus of a shell to change the target buckling strength, if the imperfections can be designed accurately before fabrication. This level of precision is difficult in practice, because imperfections can be significant during fabrication (*e.g.*, due to heterogeneity of the material, air bubbles, roughness of the casting molds, and other fabrication artifacts) and vary from one sample to another. Hence, increasing the buckling strength by increasing the thickness (or the elastic modulus, which typically increases, roughly, with density) comes at the inevitable expenses of a higher weight and may not be desirable in specific applications. Moreover, post-fabrication, the buckling strength becomes an intrinsic property determined once-and-for-all at the manufacturing stage by the material and geometrical properties. One of the central messages of our manuscript is that, through the mechanism that we uncover, one can activate the tunability of buckling for mechanical systems containing shell elements. As we have detailed before (Comment 1.1a), this could potentially open up applications in medical devices, actuators in soft robots, or metamaterials. Finally, as we will explain in more detail below (Comment 1.1d), shell buckling is a subcritical instability; hence a catastrophic event (the post-buckling path is unstable and characterized by a large drop in the pressure that can be sustained by the shell; see Fig. 1d of the manuscript). In the paper, we demonstrate how the magnetic field modifies the post-buckling behavior (the loading curves at different levels of field strength, shown in Fig. 1d), thereby limiting the catastrophic behavior of shell buckling. We emphasize that this level of control could not be achieved by simply tuning the thickness or the Young's modulus, as these intrinsic properties only affect the critical buckling pressure, but not the qualitative nature of the post-buckling behavior.

1.1.d. Fundamentally, using a magnetic field is changing the “effective modulus” of the soft materials, one can also tune the effective modulus by using polymers that respond to thermal, acoustic, light, and other stimuli.

We respectfully disagree with the Reviewer; the first part of this statement (*‘Fundamentally, using a magnetic field is changing the “effective modulus” of the soft materials...’*) is not correct. As we have clearly stated in the manuscript (centrally to the fundamental contribution that we are bringing to this class of problems), through rational arguments, we attribute the change of knockdown factor to the magnetic torque generated by the applied field due to shell rotations prior to buckling. This magnetic torque increases gradually with the shell rotation and is localized near the region where buckling occurs (see Fig. S5 in the Supplementary Note S3.4). This mechanistic interpretation was made possible by the experimentally validated theoretical model that we have developed. The fact that the magneto-elastic coupling is not merely equivalent to a change in the “effective modulus” is an important finding in our study. This nontrivial result is underlined further by the fact that the magnetic field is able to qualitatively change the nature of the buckling event, making it less catastrophic. Evidence of this statement is provided by the loading curves shown in Fig. 1d, where the slope in the linear regime (stiffness of the shell) does not change at different levels of field strength, but the value of pressure at which buckling occurs and its drop in the post-buckling

regimes varies significantly with the applied field. In other words, as we also explained in the previous point (in response to comment 1.1c), shells can be strengthened (or weakened) past the critical point less (or more) catastrophically by varying the applied field; an effect that cannot be induced by a different modulus.

In conclusion, the proposed mechanism is different from changing the “effective modulus” using other responsive materials. We hope that our clarification of this point will help the reviewer in recognizing the nontrivial aspects, and the fundamental importance, of the mechanism that we investigate.

1.2. The second question is how much the conclusion can be generalized. In most engineering applications, the shells are not a half sphere and the geometric imperfection is not a well-defined thickness at the top of the shell. Can we expect the tunability of the magnetic field valid? How far we can stretch the conclusion of this paper? I am not asking for more experiments, maybe the authors can provide more insights and interpretation of their results, considering the broad readership of Nature Communications.

In the experiments, we demonstrated the tunability of the knockdown factor for shells containing a well-defined dimple-like defect. We chose this geometric defect since it is more realistic in engineering settings compared to other types of defects, as it can be induced, for example, by bumps on the surface of the shell. For a fact, our theoretical model uncovers the basis of the underlying mechanism as rooted on the effect of the distributed magnetic torque, which is linear to the shell rotation via a stiffness that depends on the magneto-elastic parameter (Eqs. (4), (5), (S46) and (S48)). From this theory-based interpretation, we can see that the generation of the torque is not limited to a specific type of defect. Moreover, we are currently in the final stages of the preparation of a manuscript where we present a general 2D magnetic shell model, where we show that this interpretation holds also for arbitrary shell geometries. However, the level of technicality of this more recent theoretical development (a nonlinear theory for magnetic shells written in generalized curvilinear coordinates through a covariant framework) goes well beyond the scope of this paper. As such, interested readers will be drawn to our future work, for which this present manuscript sets the foundations.

Regarding the other point, our focus on a hemispherical shell does not lead to loss of generality (e.g., when compared to a full sphere or other spherical caps). The buckling of spherical shells is known to be localized at the dominant defect. Indeed, previous studies have demonstrated that shell buckling is not sensitive to the shell opening angle nor to boundary conditions (except for extremely shallow shells) [R1.8,R1.9]. Through our theoretical analysis, we have also shown that the shell rotation and the associated magnetic torque are localized in the vicinity of the defect (see Fig. S5 in Supplementary Note S3.4). This localized nature of buckling ensures that the tunability would not be affected by the shell opening angle and boundary conditions. The following sentence was added at the end of the second paragraph of the ‘Discussion’ section:

“The localized nature of buckling in spherical shells^{19,20} is also underlined in our system by the localized distribution of shell rotation and the associated magnetic torque in the vicinity of the defect (see Fig. S5 in Supplementary Note S3). This localization ensures that the tunability would not be affected by the shell opening angle and boundary conditions (except for extremely shallow shells).”

1.3. Another thing I note is that the magnetization profile is symmetric in the study. What if the shell is magnetized along the x-direction as in Figure 1c (also the Helmholtz coil is rotated 90

degrees to apply horizontal magnetic field)? For magnetic elastomer actuators, the magnetization profiles are often complex and will deform under an external magnetic field. The authors are encouraged to provide more insights into what will happen for the shells under deformation.

We very much appreciate this question and suggestion. In the earlier version of the manuscript, we had thought that demonstrating the novelty of our tunable mechanism, by way of example of the symmetric case, would have been sufficient to convey its novelty. However, motivated and inspired by the Reviewer's question, we decided to embark on a new extensive experimental campaign to explore non-axisymmetric cases (when the defect is not aligned with the magnetization and/or the applied field). Furthermore, we complemented this new experimental result with computer simulations, based on the Finite Element Method, using a recently developed user-defined element proposed in Ref. [R1.27] for hard-magnetic soft materials. This significant effort (which is at the source of why it has taken a few weeks to resubmit our manuscript) was extremely valuable as the new set of results confirms that the tunable mechanism that we proposed is still preserved in asymmetric cases. This is an important finding that further extends and strengthens the fundamental contributions of this work. Next, we provide a summary of the new results in the following, which has also been added to the main article in the "Discussion" section and in a new Figure 5 of the revised manuscript. This new material has provided a significant improvement of the overall study and we are deeply grateful to the Reviewer for instigating it through their suggestion.

In the first version of our manuscript, we had focused on axisymmetric shells and demonstrated the possibility of tuning the knockdown factor via magnetic actuation. We showed that this change of knockdown factor is dictated by the magnetic torque generated due to the shell rotation, which breaks the alignment between the magnetization (\mathbf{B}^r) and the applied field (\mathbf{B}^a). Consequently, this mechanism is expected to be preserved for asymmetric loading conditions, provided \mathbf{B}^r and \mathbf{B}^a are aligned in the initial configuration. To attest this statement, we first fabricated 16 shell specimens containing a defect at the pole with identical geometry (with average defect amplitude $\delta/h=1.64$ and defect width $\varphi_o=11.7^\circ$). The specimens were magnetized asymmetrically, at an angle φ^r with respect to the shell's axis of symmetry (see schematic diagram in Fig. R1.1a, below, which is a reproduction of the new Fig. 5a of the revised manuscript). The defect was still maintained at the pole. Then, during the buckling test, a magnetic field at the same angle ($\varphi^a=\varphi^r$) was applied, either parallel ($\mathbf{B}^r \cdot \mathbf{B}^a > 0$) or antiparallel ($\mathbf{B}^r \cdot \mathbf{B}^a < 0$) to the initial shell magnetization. In Fig. R1.1a, we plot the measured change of knockdown factor, $\Delta\kappa_d$, versus φ^r and φ^a , at $\Lambda_m=\pm 0.22$. We consistently observed finite values of $\Delta\kappa_d$ for both the axisymmetric and asymmetric cases, even when \mathbf{B}^r and \mathbf{B}^a are perpendicular to the axis of the defect. In the latter, the most unfavourable case, the range of ± 0.1 in the tuning of $\Delta\kappa_d$ is still significant. These results indicate that, even when the defect information might be unknown in an application setting, the tunable mechanism that we have uncovered can always be realized by aligning the applied field with the magnetization. Still, at the same level of Λ_m , the capacity of tuning decreases with increasing φ^r and φ^a , and the axisymmetric configuration is the most efficient at maximizing $\Delta\kappa_d$. In parallel to the experiments, we performed full 3D finite element simulations (see the "Methods" section of the paper for details) using a user-defined element proposed recently in Ref. [R1.27] for hard-magnetic soft materials. Even if the agreement between the experiments and simulations in Fig. R1.1a is excellent, we note that the FEM data exhibits a small but consistent offset from the experimental data, presumably due to the slightly different magnetic effects imposed at the shell equator (especially for the case $\varphi^a=\varphi^r=90^\circ$),

where a thick elastic band was used to hold the experimental shell specimen (Fig. 1c of the paper), in contrast to an ideal clamped boundary condition considered in FEM.

To further probe the robustness of the proposed mechanism, we considered asymmetric cases, where \mathbf{B}^r and \mathbf{B}^a were misaligned. For this case, we focused on shell specimens with a fixed magnetization set at $\varphi^r=0^\circ$, while the direction of the applied field was varied by increasing φ^a from 0° (axisymmetric, \mathbf{B}^a is parallel to \mathbf{B}^r) to 90° (\mathbf{B}^a is perpendicular to \mathbf{B}^r). The corresponding change of knockdown factor is plotted in Fig. R1.1b (same as the new Fig. 5b in the revised manuscript), whose absolute value decreases from the maximum at $\varphi^a=0^\circ$ to $\Delta\kappa_d=0$ at $\varphi^a=90^\circ$. Thus, in settings where the alignment between the shell magnetization and the external field cannot be ensured, tuning the knockdown factor is always possible (unless \mathbf{B}^a is perpendicular to \mathbf{B}^r , which leads to $\Delta\kappa_d=0$). This relaxation in loading conditions will broaden the applicability of the mechanism in real applications involving complex magnetization profiles and magnetic fields. Good agreement between FEM and experiments was found in Fig. R1.1b, which further corroborates our findings.

Fig. R1.1: **Tunability under asymmetric loading conditions.** **a**, Change of knockdown factor, $\Delta\kappa_d$, at $\Lambda_m=0.22$ ($\mathbf{B}^r \cdot \mathbf{B}^a > 0$) and $\Lambda_m=-0.22$ ($\mathbf{B}^r \cdot \mathbf{B}^a < 0$), versus angle of the shell magnetization, φ^r , and of the applied field, φ^a , which are consistently aligned in the initial configuration ($\varphi^a=\varphi^r$). In the experiments, 16 shell specimens containing a defect at the pole with identical geometry are fabricated (with average defect amplitude $\delta/h=1.64\pm 0.04$ and defect width $\varphi_o=11.7^\circ$). The shells are magnetized at an angle φ^r with respect to the axis of symmetry (2 or 3 specimens at each angle, as indicated by the number of data points). Each specimen (magnetized at φ^r) is tested with the magnetic field oriented at $\varphi^a=\varphi^r$. **b**, Change of knockdown factor, $\Delta\kappa_d$, at $\Lambda_m=0.22$ ($\mathbf{B}^r \cdot \mathbf{B}^a > 0$) and $\Lambda_m=-0.22$ ($\mathbf{B}^r \cdot \mathbf{B}^a < 0$), as a function of φ^a , with fixed $\varphi^r=0$. The two shell specimens used for the data in panel **b** were the same as those from panel **a** that were magnetized at $\varphi^r=0$.

References used in our responses to the Reviewer's comments/questions:

- [R1.1] Hilburger, M. W. Developing the next generation shell buckling design factors and technologies. In 53rd AIAA/ASME/ASCE/AHS/ASC Structures, Structural Dynamics and Materials Conference, Structures, Structural Dynamics, and Materials and Co-located Conferences (American Institute of Aeronautics and Astronautics, Honolulu, HI, 2012).
- [R1.2] Tsien, H.-S. A theory for the buckling of thin shells. *Journal of the Aeronautical Sciences* **9**, 373-384 (1942).
- [R1.3] Hutchinson, J. W. Imperfection sensitivity of externally pressurized spherical shells. *J. Appl. Mech.* **34**, 49-55 (1967).

- [R1.4] Carlson, R. L., Sendelbeck, R. L. & Hoff, N. J. Experimental studies of the buckling of complete spherical shells. *Exp. Mech.* **7**, 281-288 (1967).
- [R1.5] Lee, A. et al. Fabrication of slender elastic shells by the coating of curved surfaces. *Nat. Commun.* **7**, 11155 (2016).
- [R1.6] Lee, A., Lopez Jimenez, F., Marthelot, J., Hutchinson, J. W. & Reis, P. M. The geometric role of precisely engineered imperfections on the critical buckling load of spherical elastic shells. *J. Appl. Mech.* **83**, 111005 (2016).
- [R1.7] Hutchinson, J. W. EML Webinar overview: New developments in shell stability. *Extreme Mech. Lett.* **39**, 100805 (2020).
- [R1.8] Hutchinson, J. W. Buckling of spherical shells revisited. *Proc. R. Soc. A* **472**, 20160577 (2016).
- [R1.9] Hutchinson, J. W. & Thompson, J. M. T. Nonlinear buckling behaviour of spherical shells: barriers and symmetry-breaking dimples. *Phil. Trans. R. Soc. A* **375**, 20160154 (2017).
- [R1.10] Yan, D., Pezzulla, M. & Reis, P. M. Buckling of pressurized spherical shells containing a through-thickness defect. *J. Mech. Phys. Solids* **138**, 103923 (2020).
- [R1.11] Virost, E., Kreilos, T., Schneider, T. M. & Rubinstein, S. M. Stability landscape of shell buckling. *Phys. Rev. Lett.* **119**, 224101 (2017).
- [R1.12] Djellouli, A., Marmottant, P., Djeridi, H., Quilliet, C. & Couplier, G. Buckling instability causes inertial thrust for spherical swimmers at all scales. *Phys. Rev. Lett.* **119**, 224501 (2017).
- [R1.13] Whitesides, G. M. Soft robotics. *Angewandte Chemie Int. Ed.* **57** (16), 4258-4273 (2018).
- [R1.14] Reis, P. M. A perspective on the revival of structural (in)stability with novel opportunities for function: from buckliphobia to buckliphilia. *J. Appl. Mech.* **82**, 111001 (2015).
- [R1.15] Kochmann, D.M. & Bertoldi, K. Exploiting microstructural instabilities in solids and structures: from metamaterials to structural transitions. *Appl. Mech. Rev.* **69**, 050801 (2017).
- [R1.16] Holmes, D. P. Elasticity and stability of shape-shifting structures. *Curr. Opin. Colloid Interface Sci.* **40**, 118-137 (2019).
- [R1.17] Gorissen, B., Melancon, D., Vasios N., Torbati, M. & Bertoldi, K. Inflatable soft jumper inspired by shell snapping. *Science Robotics* **5**, eabb1967 (2020).
- [R1.18] Jampani, V. S. R., Mulder, D. J., Sousa, K. R. D., Gilbert, A.-H., Lagerwall, J. P. F. & Schenning, A. P. H. J. Micrometer-scale porous buckling shell actuators based on liquid crystal networks. *Adv. Funct. Mater.* **28**, 1801209 (2018).
- [R1.19] Datta, S. S. , Kim, S.-H. , Paulose, J. , Abbaspourrad, A. , Nelson, D. R. & Weitz, D. A. Delayed buckling and guided folding of inhomogeneous capsules. *Phys. Rev. Lett.* **109**, 134302 (2012).
- [R1.20] Vian, A. & Amstad, E. Mechano-responsive microcapsules with uniform thin shells. *Soft Matter* **15** (6), 1290-1296 (2019).
- [R1.21] Sacanna, S., Irvine, W. T. M., Rossi, L. & Pine, D. J. Lock and key colloids through polymerization-induced buckling of monodisperse silicon oil droplets. *Soft Matter* **7**, 1631-1634 (2011).
- [R1.22] Hu, W., Lum, G. Z., Mastrangeli, M. & Sitti, M. Small-scale soft-bodied robot with multimodal locomotion. *Nature* **554**, 81-85 (2018).
- [R1.23] Kim, Y., Yuk, H., Zhao, R., Chester, S. A. & Zhao, X. Printing ferromagnetic domains for untethered fast-transforming soft materials. *Nature* **558**, 274 (2018).
- [R1.24] Kim, Y., Parada, G. A., Liu, S. & Zhao, X. Ferromagnetic soft continuum robots. *Sci. Robot.* **4**, eaax7329 (2019).
- [R1.25] Alapan, Y., Karacakol, A. C., Guzelhan, S. N., Isik, I. & Sitti, M. Reprogrammable shape morphing of magnetic soft machines. *Sci. Adv.* **6**, eabc6414 (2020).
- [R1.26] Gu, H. et al. Magnetic cilia carpets with programmable metachronal waves. *Nat. Commun.* **11**, 2637 (2020).
- [R1.27] Zhao, R., Kim, Y., Chester, S. A., Sharma, P. & Zhao, X. Mechanics of hard-magnetic soft materials. *J. Mech. Phys. Solids* **124**, 244-263 (2019).

2. Response to Reviewer #2

The stated purpose of this manuscript is

Here, we propose a robust mechanism to dynamically tune the buckling strength of shells (i.e., their knock-down factor), by coupling elastic deformation and magnetic actuation, to provide active control and liberate the predestination of imperfections.

Toward that, the authors have shown that the buckling strength of shells can be dynamically tuned by exploiting the coupling between mechanics and magnetism. The authors used a combination of bench scale precision experiments and theoretical modeling on hard-magnetic elastomeric shells. They show that the tuning mechanism is a distributed torque induced by the magnetic field throughout the shell. And they also show that a dimensionless quantity, the magneto-elastic buckling number, emerges as the key governing parameter, summarizing the geometrical, mechanical, and magnetic properties of the system. The results are novel, and showcase how buckling may be tuned using applied magnetic fields. Overall the manuscript is in an excellent condition and only needs to consider the following minor comments:

We are grateful for the Reviewer's encouraging and supportive feedback and recognition of the novelty and quality of our work.

2.1. In S1.2, the authors state:

We used a simple model to provide an analytical description of the geometry of the defect. Note that, during pressurization of the mold, the bulk region outside of the soft spot exhibits negligible deformation compared to the soft spot. Moreover, the spherical cap of this soft spot is sufficiently small (with respect to the radius of curvature of the shell), such that it can be regarded as a (nearly) flat plate. Hence, we isolate this thin region and model it as a boundary-clamped circular flat plate loaded by a uniform pressure.

Is there any possible quantification to help support the (reasonable) assumption for the clamped boundary condition?

We note that the measured angular width of the defects produced in the MRE-22 shells is 11.7° , which is larger than that of the thin region of the mold (which is 10°). This small but finite difference indicates that the boundary of the thin region was not ideally clamped. However, the purpose of the solution obtained from the boundary-clamped plate model is to offer an appropriate qualitative description of the measured profile, as long as the defect width and amplitude are measured by fitting the theoretical deflection to the experimentally measured defect profiles. As such, the plate model serves only as a descriptor of the defect, for example to then include the exact same defect profile (with a well-posed functional form) into the magnetic shell theory or the (new) Finite Element simulations. The excellent agreement shown in Fig. 2b of the manuscript between the measured experimental profile (obtained through optical profilometry) and the clamped plate model provides evidence that this descriptor is indeed appropriate.

We have clarified this point in the Supplementary Note S1.2 of the revised manuscript:

“Although the boundary of the thin region was constrained by the flexible bulk region, we found the measure profiles can be well described by the solution corresponding to a clamped boundary

condition (see Fig. 2b of the main article). However, the amplitude and width of the defect have to be determined through fitting (see details below)."

To answer the Reviewer's question more specifically, we can look at our defects as very shallow shells as follows. A shallow shell is defined, as a rule of thumb, as a shell whose half angle is smaller than around 30° [R2.1]. As our defects cover an area whose half angle is at most 11.7°, we can look at our defects as very shallow shells or plates. Still, since as we say above we are only using the clamped plate solution as a descriptor of the defect profile, we believe that it is not necessary to provide this level of quantification in the manuscript.

[R2.1] Reissner, E. Stresses and small displacements of shallow spherical shells. I. *Journal of Mathematics and Physics* **25**, 80-85 (1946).

2.2 In S2.2, the second paragraph under "Protocol of the buckling test" states

Noting that the flow rate of 0.4mL/min for the pressurization was set constant for all experiments, these experimental could run for up to 70s, which was far too long for the coils to be switched on.

but the word experimental is most probably intended to be experiments.

This was indeed a typo, which we fixed; thank you.

2.3 In S2.2, the second paragraph under "Protocol of the buckling test.' Can some description be added to the heating of the pressurized air as a motivation for not continuously applying the magnetic field. I do not know much about these coils, and why they would heat up the air in the pneumatic system.

During the experiments, the temperature in the coils (which are made of copper wire, acting as a resistor) increases due to Joule heating; passage of electric current through the wire produces heat. The power of heating generated is proportional to the product of the wire's Ohmic resistance and the square of the current flowing through it (this is the classic Joule–Lenz law). Consequently, this Joule heat is dissipated to the surrounding air through a combination of convection and radiation. Since the shell specimen connected to the pneumatic loading system was placed in between the two coils, the air inside the shell can also heat up to an undesirable level, if the coils are left on for an excessively long time.

In the pneumatic system, the number of air molecules was constant (n_i) and the syringe was expanding the air volume (V_i). First, we consider the case of our experiments where the temperature remained nearly constant (T_i). According to the ideal gas law, $p_i V_i = n_i R_i T_i$, where R_i is the gas constant. The inner pressure of the shell $p_i = n_i R_i T_i / V_i$ was decreased due to the increase of V_i by the syringe. Since the ambient pressure (p_o) was constant, the shell was loaded by the pressure difference $p_i - p_o$, which was negative in our experiments, *i.e.*, the shell was pressurized by the ambient air.

If the air inside the pneumatic system heats up due to the Joule heating of the coil, the change of inner pressure $p_i = n_i R_i T_i / V_i$ would depend on the ratio T_i / V_i . Since both T_i and V_i increase in the pneumatic system, it is hard to determine whether the inner pressure p_i would increase or decrease. We observed in our preliminary experiments that, when the coils overheated (T_i quickly increased), a situation which we tried to avoid, by expanding the air at the same rate (flow rate

controlled by the syringe pump), the decrease of inner pressure p_i became slower and, eventually, p_i increased. As a result, the pneumatic system could not generate increasing pressure loading to the shell.

To address this point, we have provided the following information in the Supplementary Note S2.2 of the revised manuscript:

“Avoiding overheating of the coils (due to Joule heating) during the experimental tests was crucial. Otherwise, the decrease of inner pressure of the shell p_i (by expanding the air volume V_i using the syringe pump) would be compromised by the increasing temperature T_i of the heated air inside the shell, according to the ideal gas law $p_i = n_i R_i T_i / V_i$ (n_i is the constant number of air molecules in the system and R_i is the gas constant). As a result of overheating, the pressure loading applied on the shell would have decreased before reaching the critical value.”

3. Response to Reviewer #3

This paper presents a new means of controlling the buckling threshold of shells subjected to an external pressure. The idea is that shells with tunable mechanical response can be made by adding hard magnetic particles to the polymer mixture used to form a shell - these particles have an intrinsic magnetic moment after an initial magnetisation step, which is referred to as a residual magnetic field in this paper. When the composite structure is then subject an external pressurization then the response of the shell can, to an extent be tuned by the application of an additional field (B^a). The authors demonstrate this by considering the threshold pressure required for buckling as a function of the various parameters in the problem. In particular, they show that this buckling is imperfection-sensitive (a result that is well known from the literature without a magnetic field), and characterize this by measuring a knock-down factor. They present experimental and theoretical results that are in good agreement, showing that they have developed a good understanding of the underlying phenomenon.

We appreciate the Reviewer's positive feedback and helpful suggestions, all of which have been carefully considered and implemented. In particular, we have thoroughly read the past work pointed out by the Reviewer and revised the introduction of the paper to further highlight the novelty of our work. We have also added more details of the scaling analysis to clarify our argument. In the final conclusion paragraph of the revised manuscript, we emphasized the tunable nature of our mechanism and pointed to its potential for applications in future designs of functional devices with tunable mechanical properties and behavior.

3.1 General comments: I have some technical questions, which I describe below, but my general sense is that the paper is really more suited to a specialist mechanics journal, rather than a general interest one like Nature Communications. I think the idea is a really neat one, but, for the moment, the authors did not convince me that they have a 'killer app' that could be developed with this idea - indeed, the paper finishes a little weakly with the authors supposing that "magnetic structures with predictable and tunable buckling behavior will enrich the design of advanced materials with new functionalities" but do not give more details. This asks follow-up questions: what new functionalities would this enable, and how can this be done when the maximum relative change in knock-down factor is about 50%?

We understand the Reviewer's concern and we want to provide here the rationale why we believe that our work goes beyond the examples illustrated in the paper, both from the perspective of fundamental modeling and applications. Shell buckling is a fundamental and longstanding classic problem in structural mechanics [R3.1-R3.3], where shell structures are inevitably imperfect due to manufacturing or as a cause of continued usage. When studying this phenomenon, it is customary (and historical) to focus on one or two prototypical examples, namely the buckling of spherical or cylindrical shells, which serve as bases for analyses of more complex shell-like systems. Indeed, this methodological focus is justified by the understanding that an analysis of arbitrarily curved shell structures could only be carried out via finite element analysis (treated as a 'black box') with conclusions that would be case-specific rather than general. By contrast, understanding how positively curved (spherical) or flat (cylindrical) shells respond to loads has enriched our understanding of shell buckling as a general process, and informed the design of more complex structures by establishing fundamental design guidelines and scaling laws.

It is important to highlight that in introducing and solving the present problem of the pressure buckling of a magnetically active spherical shell, we have fabricated and modeled, for the first time, initially curved hard MREs thin structures. As a consequence of our study, we uncovered the novel mechanism that a shell's knockdown factor can be dynamically tuned, on-demand. In addition to, quoting the reviewer, this being '*a really neat idea in the field of shell buckling*', our study opens exciting new avenues that go well beyond shell buckling itself. Specifically, while rationalizing and developing a predictive understanding for our problem, we have, for the first time:

- i) Fabricated and demonstrated the working principles of initially curved magneto-active thin structures that can be actuated remotely by a magnetic field;
- ii) Developed a fully predictive reduced model, with no fitting parameters, that is able to quantify the variation in knockdown factor. More broadly, our magnetic shell theory is also able to model the large deformations of such magnetic structures in more complex contexts where shell buckling might not be even present.

We believe that the combination of both of these points is an important contribution that will pave the way for future designs of original magneto-active shell structures in a variety of application settings. Our advance in augmenting recent pioneering research on hard-magnetic MREs [R3.4-R3.7] to now also being able to model initially double-curved structures is far from trivial and decidedly not a mere theoretical extension of what was already present in the literature. Our framework will be an important step for the design of future actuators since initial curvature brings in the possibility of tuning and controlling an important class of other elastic instabilities: snapping; that is a limit-point instability. This instability type will equip future magneto-active systems with another '*knob*' to enhance large deformations, and shape change, taking advantage of multiple stable states, as shown recently for robotic systems [R3.8]. In other words, even though our model was developed to understand the pressure buckling of magneto-active spherical shells, we have since realized that the framework we introduced is actually a far more general tool that can be used to understand and design initially curved structures undergoing large deformations. We are in the process of drafting a more technical manuscript on this topic, which we intend to submit to a specialized journal, where we develop a reduced theory for non-axisymmetric shells undergoing non-axisymmetric deformations. This model will allow fast computations of non-trivial deformations undergone by magnetic shells, therefore being a fundamental tool for the design of these structures.

Finally, we want to emphasize that even in the specific case of shell buckling that we presented in the paper, the variation of knockdown factor can be made to be larger than the $\pm 30\%$ range that we reported. Indeed, this is an upper bound specific to our experimental setup and not a fundamental limit of the general phenomenon. For example, if shells were softer or the applied magnetic field was stronger, the variations in knockdown factor would be even larger than what we observed in our work, since the magneto-elastic parameter will be larger.

As we also mentioned in our reply to Reviewer 1, future developments of '*killer apps*' (the absence of which we do not believe should constitute a filter for a broad and high-profile journal such as *Nature Communications*) can only be made possible if engineering design advances hand-in-hand with more fundamental efforts, as is the case of our manuscript. In fact, we identified *Nature Communications* as the ideal venue for our manuscript precisely for its broad range in reporting

studies along the full fundamental-to-applied spectrum (and not just the latter). Our contribution is decidedly towards the former, under the umbrella of engineering science/mechanics.

Still, motivated by the Reviewer's comment, we have added the following sentence in the final paragraph of the 'Discussion' section (just before 'Methods'):

"We hope that the principle that we have uncovered will open an axis of tunability in the design of mechanical systems where shell buckling is harnessed as a functional mechanism ^[56,57,58], to offer devices with tunable mechanical properties or behavior."

3.2 Previous work: There is a large body of work on two problems that are closely related to that considered here: buckling of elastic objects driven by magnetic fields (magneto-elastic buckling) and the magnetism induced resistance to buckling under other external fields (but no applied magnetic field). At some level, the current paper seeks to combine these two points of view. However, as far as I could tell, this paper misses the main literature in each of these areas. I think it would be appropriate to consider these literatures, and to cite the relevant works appropriately.

(i) There is a long history of work on buckling induced by magnetic fields; I believe the classic work is by Moon and Pao, Appl Mech 36, 92 (1968) but there has also been a considerable amount of recent work on this problem ongoing, see Gerbal Proc. Natl Acad Sci 112, 7135 (2015) and Cebers, Phys Rev E 76, 031504 (2007).

There have also been many studies on the properties of chains of paramagnetic particles, which have an effective tension within the chain because the induced magnetic moments are always aligned with the applied field (see, for example, the work of Roper, Dreyfus et al: Nature 436, 862 (2005) and especially section 2.2 of J. Fluid Mech. 554, 167 (2006), as well as the work of Cebers, e.g. Phys Rev E 70 021404 (2004)).

(ii) The second related problem concerns the properties of chains of ferromagnetic beads. Qualitatively speaking, these have an effective bending stiffness arising from the permanent magnetic dipole, which stiffen such chains from buckling under the influence of gravitational loading (see, for example, work by the groups of Goriely, Proc R Soc A 470 20130609 (2014) and Fried, Proc R Soc A 473 20160703 (2017)). If I understood these works properly, they seem to have shown that there's an effective Young's modulus E_{eff} proportional to B^2/μ_0 where B is the magnetic field strength of the beads.

In the framework of elastic (non-magnetic) structures, the mechanics of shells is dramatically different from that of beams, filaments, or elastica, which is why we had decided to place the emphasis of our literature review on shell systems (on which we focus exclusively throughout the manuscript) and the importance/significance of predicting/tuning their knockdown factor. With that said, as pointed out by the Reviewer, we recognize that our original literature review may come across as incomplete from the perspective of past work on the magneto-elastic coupling. We agree with the Reviewer that it will be appropriate and relevant to more thoroughly acknowledge previous studies where magneto-elastic systems were investigated, which we have addressed in the revised version of our manuscript (more details on this below). Still, in the paragraphs below, we provide a classification and overview of the wide range of possible different material systems that exhibit magneto-elastic coupling. In doing so, we seek to better clarify the context of novelty of our work, not only from the perspective of tunable buckling strength, but also in our usage of recent developments in hard-ferromagnetic soft materials. In fact, we believe that our changes in

the revised manuscript to better reflect past work on magneto-elastic systems have helped us in more clearly stating the niche of our work, and we are grateful to the Reviewer for the suggestion. We stand by the statement of novelty of our work on magnetic shells, for which we have uncovered a novel mechanism for tuning their buckling strength.

First, we believe that it may be helpful to clarify the different possible regimes of magneto-elastic materials, which are then detailed below:

- (i) Superparamagnetic and soft-ferromagnetic materials;
- (ii) Hard-ferromagnetic materials;
- (iii) Chains of hard-ferromagnetic spheres;
- (iv) Magnetorheological elastomers (MREs) embedded with soft-ferromagnetic particles; and
- (v) Magnetorheological elastomers (MREs) embedded with hard-ferromagnetic particles (which we use in the present manuscript).

(i) *Superparamagnetic or soft-ferromagnetic materials* have a large magnetic susceptibility. Hence, such materials can be magnetized, albeit temporarily, by an external magnetic field. Indeed, upon removal of the field, the magnetization of superparamagnetic materials “vanishes” (on average) to zero. Moreover, soft-ferromagnetic materials can retain the magnetization (partly). However, due to their low coercivity, they can be easily demagnetized (or remagnetized) by another field. In short, the magnetization of superparamagnetic and soft-ferromagnetic materials depends strongly on the external field. Structural elements made out of these materials, when subjected to a uniform field, can be deformed as a result of the magnetic torques generated when the field and its induced magnetization are not aligned [R3.9-R3.15].

(ii) *Hard-ferromagnetic materials* possess sufficiently high remanent magnetization and coercivity upon saturation. When a structure made of hard-ferromagnetic material is loaded under a magnetic field (lower than the coercivity, which can be satisfied in most applications), the field can not change the remanent magnetization, which, in other words, is independent of the external field. By applying a uniform magnetic field at an angle to the material magnetization, one can generate magnetic torques to actuate the structure.

(iii) *Chains of hard-ferromagnetic spheres* are pure magnetic systems without elasticity [R3.16,R3.17]. The internal forces that balance external mechanical loading stem from the magnetic interaction between the spheres, exhibiting a variety of equilibrium configurations. These magnetic systems can be interpreted as beams with an ‘effective’ bending stiffness, even if this stiffness is not elastic in nature and is determined solely by the magnetic properties of the spheres.

(iv) *Magnetorheological elastomers (MREs) embedded with soft-ferromagnetic particles* can deform under an external magnetic field via magneto-elastic coupling [R3.18-R3.24]. Driven by the magnetic field, the distributed magnetic particles tend to align themselves with the field, thereby forming particle chains, which involves elastic deformation in the matrix. These microscopic movements of the particles in the soft matrix result in the macroscale “magnetostriction” of the composite. Also, the re-arrangement of the particles and the particle interaction change the effective modulus of the composite, in the presence of a magnetic field. To enhance the magneto-mechanical coupling, a magnetic field is usually applied to MREs during the curing process, such that particles could be arranged in a roughly column-like structure.

(v) *Magnetorheological elastomers (MREs) embedded with hard-ferromagnetic particles* [R3.4-R3.7, R3.25-R3.28] inherit their hard-magnetic properties from the particles. Thus, the composite is magnetically hard and retains the magnetization upon saturation. Due to the soft elastomeric matrix, the composite is mechanically compliant and, as such, it can exhibit significant shape changes in response to an external field. This fast and reversible shape-morphing of hard MREs, which can be controlled remotely by an external magnetic field, has been exploited for functional devices in several recent applications. Examples include soft robotics [R3.5,R3.25-R3.27] and micromachines [R3.28]. In these systems, by locally orienting the magnetization of the particles, one can program the magnetization profile of the material [R3.4,R3.5,R3.28], which enables to generate complex deformations/shapes via a sophisticated magnetization profile.

The works [R3.9-R3.15] mentioned by the Reviewer all focus on superparamagnetic or soft-ferromagnetic rods or chains, classified in (i) above. At this stage, it is important to highlight that these material systems are different from the hard-ferromagnetic elastomers, which we have used in our systems, classified as (v) above. Back to the papers mentioned by the Reviewer, these rod-like systems have a magnetization that is induced by an external field and their mechanical deformation can be modeled by considering an appropriate bending stiffness. Due to the slenderness of the rod, this magnetization is almost entirely along the beam axis for fields in any direction (unless the field is perpendicular to the rod's centerline). This misalignment between the applied field and the induced magnetization generates magnetic torques that deform the rod. From Eq. (1) of Ref. [R3.11], the magnetic energy considered in this paper is proportional to H_0^2 , where H_0 is the field strength of the external field. When the field is applied perpendicularly to the rod, the magnetization is originally along the transverse direction. Once the field strength reaches a critical value, the magnetization becomes mostly axial and the rod suddenly deforms under the generated magnetic torques and buckling occurs [R3.9,R3.15].

By contrast, in our work, we consider shells made of an elastomer embedded with hard-ferromagnetic particles, as the Reviewer correctly identifies at the start of the comment (but the mentioned references points to a possible confusion). Indeed, as we have mentioned in the above classification for (i) and (v), the actuation mechanism of our system is different from that made of superparamagnetic or soft-ferromagnetic materials: the magnetization of our shell is permanently encoded in the material (through the saturated hard-ferromagnetic particles), independently of the external field applied during actuation. As such, in our system, the magnetic energy is proportional to $B^r B^a$ (Eq. (3) in the manuscript), where B^r is the residual flux density (magnetization), which is independent of the applied external flux density B^a (field strength). This feature enables us to programme sophisticated magnetization profiles on structures, thus allowing for design with complex shape changes via magnetic actuation. We will highlight this recent development later.

Moving on to the two other papers mentioned by the Reviewer, Vella *et al.* [R3.16] and Schönke & Fried [R3.17] focused on chains of hard-ferromagnetic spheres, which were classified in (iii) above. These are pure magnetic systems (void of elasticity), which are self-assembled through internal magnetic interactions. The internal magnetic forces induced by the relative motion of the spheres can resist external mechanical loading. From a modeling perspective, Refs. [R3.16] and [R3.17] regarded this effect as analogous to beam bending, thus considering an effective bending stiffness. This effective bending stiffness is taken to be proportional to B^2/μ_0 (Eq. (2.25) in Ref. [R3.16]), where B is the magnetic field strength of one sphere.

Again, by contrast, in our shell system, we have used a magneto-elastic system, involving both magnetism and elasticity. The magnetic effect that couples with shell rotations arises from the interaction between the hard-magnetic material and the externally applied field, rather than the interactions between hard-magnetic particles within the shell. In our response to the technical question (1) addressed below, we have demonstrated that the particle interactions in our shells are too weak to change the buckling load. Therefore, the nature of the mechanisms in the two systems (ours and that studied by Vella *et al.* [R3.16] and Schönke & Fried [R3.17]) are fundamentally different. In our answer to the technical question (4), below, we detail that the magneto-elastic coupling in our system is far from being equivalent to a change in the “effective modulus”. This interpretation by the reviewer was not correct.

We re-emphasize that, in our study, we have focused on shells made of hard-magnetic materials, for which a predictive mechanical understanding was only developed very recently [R3.5,R3.6]. The modeling of this class of materials is based on the Helmholtz energy of the system, which includes both elastic and magnetic energy components [R3.6]. The latter is coupled with deformation, accounting for the magnetic effect arising from the interaction between the material magnetization and the external field. Starting from this energy formulation, the pioneering work in Refs. [R3.5, R3.6] implemented a constitutive description of these hard-magnetic materials to enable full 3D finite element modeling (FEM), showing the ability to quantitatively predict the behavior of structures made out of hard-magnetic soft materials. Still, to date, structural theories for hard-magnetic thin structures are limited to simple 1D structural elements (linear beams and elastica) [R3.4,R3.7]. These recent advances at the level of a constitutive description of hard-magnetic soft materials, coupled with the more complex structural layouts that we demonstrate for a shell system, could provide a large design space for shells in various applications, where the magnetization profile needs to be programmed. As we have mentioned above the magnetization of hard-magnetic soft materials is independent of external magnetic fields. Based on this principle, recent pioneering developments have demonstrated the ability to realize complex and tunable deformations/locomotions of magneto-elastic systems via magnetic actuation, attributed to hard-magnetic materials with programmable remanent magnetization [R3.4,R3.5,R3.28]. For example, many promising applications are starting to be realized in soft-robotics [R3.5,R3.25,R3.27,R3.28] and invasive medicine [R3.26]. Next, we briefly overview several recent developments in this area. For example, Lum *et al.* [R3.4] proposed to programme the magnetization profile of a beam by bending/folding the beam via laser-cut jigs during the magnetization process, where the orientation of magnetization at a material point is related to the deformation. The magnitude of magnetization was adjusted by varying the particle concentration along the beam axis. Using this technique, given a desired shape/deformation of the beam, they were able to realize it by programming the magnetization profile. Also benefiting from this technique, Hu *et al.* [R3.25] designed magneto-elastic soft robots with multimodal locomotion. In another example, Kim *et al.* [R3.5] reported an approach to design and 3D print (direct ink writing) shape-programmable soft materials with ferromagnetic domains. The magnetized hard-ferromagnetic particles in the ink were re-oriented during printing, according to the designed domain pattern. Under a uniform magnetic field, structures with programmed ferromagnetic domains were capable of fast transformations between complex 3D shapes.

In summary, we agree with the Reviewer that it is appropriate to mention past work more thoroughly, which we now see as actually helping us in further highlighting the novelty of our own work. In the last paragraph of the introduction, we have added several sentences that first mention the pioneering work on superparamagnetic and soft-ferromagnetic systems and then

contrast them to more recent advances in (and the unique features of) hard-magnetic materials, with which we highlight the novelty and achievement of the work on shells:

“Here, we propose a robust mechanism to dynamically tune the buckling strength of shells (i.e., their knockdown factor), by coupling elastic deformation and magnetic actuation, to provide active control and liberate the predestination of imperfections. Past pioneering studies have addressed the deformation and buckling of magneto-elastic systems made of superparamagnetic^[30-36] or soft-ferromagnetic^[37-43] materials under an external magnetic field. More recent advances have tended to focus on hard-magnetic soft materials^[44-47], which are magnetorheological elastomers (MREs) embedded with hard-ferromagnetic particles. This class of materials is magnetically hard with programmable remanent magnetization and high coercivity upon saturation, but mechanically compliant due to the soft elastomeric matrix. These characteristics enable fast, reversible, and complex shape-morphing through remote magnetic actuation, as has been exploited for functional devices in a variety of applications, including shape-programmable materials^[44,45], biomedical devices^[48], and soft robots^[49-51]. Our framework leverages the unique features of hard-magnetic soft materials in the context of shell mechanics.”

Furthermore, when we introduce the magnetic energy of hard-magnetic materials in subsection ‘Theory of axisymmetric hard-magnetic shells’ of the revised version of the manuscript, we now clearly state the difference in energy between the hard-magnetic material used in this work and superparamagnetic/soft-ferromagnetic material considered in previous work:

“We note that, in the magnetic energy of hard-magnetic materials, \mathbf{B}^r is the remanent magnetization retained in the material (through the saturated hard-ferromagnetic particles), which is independent of the external field \mathbf{B}^a . This independence between \mathbf{B}^r and \mathbf{B}^a in hard-magnetic materials contrasts with superparamagnetic and soft-ferromagnetic systems^[31-33,36,42,43], where the material magnetization is induced by the field applied for actuation, thereby relating to \mathbf{B}^a .”

3.3 Technical questions:

(1) I am confused about why the magnetic energy of eqn (S4) includes only the interaction between the residual field and that of the applied field? Based on the papers mentioned in (ii) above, there should also be an energetic cost associated with the deformation of the shell because of the interaction of the residual magnetic field with itself (or rather other parts of itself). I originally thought that this could be neglected if $B^r \ll B^a$, but it seems that this is not the case experimentally. Could the authors discuss this point.

As mentioned in our reply to Comment 3.2, the hard-magnetic elastomeric material that we used in our study is different from the superparamagnetic or soft-ferromagnetic materials considered in the references brought up by the Reviewer. Consequently, the relevant expressions to describe magnetic energy of either material are also different. As pointed out above, in the works by Vella *et al.* [R3.16] and Schönke & Fried [R3.17] mentioned by the Reviewer, the authors investigated the bending rigidity stemmed from the strong interaction between the magnetic spheres, which were stacked in a single chain with the north pole of one ball connecting to the south pole of another pole.

In our system, the effect of the interaction between magnetized particles (interaction of the residual magnetic field with itself) depends not only on the magnitude of magnetization B^r , but also on the concentration and arrangement of the particles as well as the shape of the structure (the distance between different parts). In the material that we have used to fabricate our shells, the volume

concentration of magnetic particles is 7% and the particles were distributed randomly in the silicone polymer matrix (no magnetic field was applied at curing), thus no particle chains formed. We anticipate that the particle interactions were weak compared to the interaction between the applied field and the particles. Moreover, prior to buckling of the shell, different regions of the shell are distanced without contact, such that interactions between these different regions should also be weak.

Motivated by the Reviewer's question, toward evaluating the influence of particle interaction in the geometry and buckling strength of our shells, we have performed a new set of experiments to measure the defect amplitude and the critical buckling pressure of shells before and after magnetizing. We considered imperfect shells with small and large defects (see Fig. R3.1a, below). In these new experiments, differently from the fabrication protocol described in the paper, we peeled off the non-magnetized shell from the mold before magnetizing and measured the defect amplitude and the critical buckling pressure. We then magnetized the shell and did the measurements again on the magnetized shell. Throughout the buckling tests, we did not apply any external field. The only difference between the two sets of experiments was the presence (or absence) of the residual magnetic field in the magnetized (or non-magnetized) shell.

Fig. R3.1: **a**, Defect amplitudes and **b**, knockdown factors of shell specimens before and after magnetizing. The nearly identical results demonstrate that the effects of self-interactions of the residual magnetic field are not important in our system.

From the results shown in Fig. R3.1, we conclude that the defect amplitude (Fig. R3.1a) and knockdown factor (Fig. R3.1b) were approximately the same before and after magnetizing. This finding indicates that the residual magnetic field or particle interaction in our shell system was too weak to affect the results. This is also why we only considered the interaction between the remanent magnetization and the applied field in our shell model and obtained an excellent quantitative agreement with the experiments, with no fitting parameters. These new results, together with the excellent experiments/theory agreement that we had already demonstrated in the manuscript, confirm that the simplified model is sufficient to describe the buckling of our shells. We agree that more complex settings, where self-interactions of the residual magnetic field may play a role, will require more sophisticated modeling approaches, although this is not necessary here and such a treatment goes beyond the scope of our current study.

(2) One of the main results of the paper is the scaling in eqn (6). However, I was not able to understand the scaling analysis that is meant to lead to this result, which is given in section 3 of the SI. For example, in (S51) a complex integral with factors of λ_m/R^4 is comprehensible but is immediately replaced in scaling terms by $\lambda_m h$ after

apparently “noticing a linear proportionality to the thickness-to-radius ratio”. What exactly does this mean (which terms scale with thickness etc? What happens to the extra factor of R^3 that was in the first portion of (S51)?). I have similar comments/questions about the other scaling laws in this section. Others, I do not understand at all. For example, after (S53) we are told that the volume scales as $R^2 h$. This is not what I would expect for pressure buckling, where the radial scale of the dimple that forms scales as $(Rh)^{1/2}$ and has height h (hence volume $h^2 R$). What did I miss?

We recognize that the scaling analysis, as presented in Supplementary Note S3.3, could have been explained more clearly. Motivated by the Reviewer’s comment, we have modified the text in Supplementary Note S3.3 so as to better clarify our scaling argument. In particular, the energy term in Eq. (S51) is explained in detail with the addition of the following text:

“This scaling can be derived as follows. The first term within the square brackets scales as Rh , given that the function $\psi(\varphi)$ is dimensionless, whereas $b_{11} \sim R$ and $w \sim h$. The second term also scales as Rh , given that the term within the round brackets is dimensionless and $u_1 \sim Rh$ (the magnitude of the in-plane displacement field scales as h but the covariant components scale as Rh since the contravariant base vectors scale as $1/R$). By performing the required multiplications, remembering also that $a \sim R^2$, it follows that $U_m^{(1)} \sim \lambda_m h/R$.”

Note that the prefactor R^2 is canceled out by the area measure, which scales as R^2 .

Regarding the energy term in Eq. (S52), we highlight that the function $\hat{\chi}$ is dimensionless. In this way, the scaling of this energy term is clearer, since it is dictated by the scaling of the rotation vector.

The energy term in Eq. (S53) is admittedly the most cumbersome one, for the complexity of the term itself, to which our original presentation of the scaling argument did not help. To make the presentation more accessible, we now refer to Eq. (S49), where the distributed torque $\mathbf{m}(\mathbf{u})$ is defined, and perform the scaling analysis for this term first. To achieve this, we use all the observations used for the previous terms. We added the following text to explain our analysis:

“The scaling for this term can be derived by applying the same arguments that we used for $U_m^{(2)A}$, for which we first turn to the second line of Eq. (S49). The first term of the multiplication scales as $1/R^3$. The second term, that is, the covariant component of the rotation vector, scales as $(Rh)^{1/2}$ as we have seen previously. Finally, the third term in the square brackets scales as Rh , exactly as we have seen for the first-order energy term. Multiplying all these terms, we conclude that the second-order term $U_m^{(2)B}$ scales as $\lambda_m (h/R)^{3/2}$.”

Finally, the reviewer is concerned with the scaling of the change in volume, which leads to the scaling of the live pressure potential, namely Eq. (S55). The scaling that the reviewer suggests, $\Delta V \sim Rh^2$, is correct in the post-buckling regime, where a dimple has formed. However, buckling occurs well before the formation of the dimple, and we are required to focus our scaling analysis on the energy levels before and up to buckling (*i.e.*, in the linear regime). In this case, the shell is (almost) uniformly compressed in a smaller spherical shape where only the normal displacement is nonzero and of the order of the thickness. In this case, as we now demonstrate more explicitly in the SI, the variation of the volume scales as $R^2 h$. Since all volume-related quantities are then normalized by $R^2 h$, the dimensionless variation in volume is of order 1. We have added the following text in the SI to reflect this comment:

“In the pre-buckling regime (during loading up to the buckling event), the displacement field is normal to the middle surface of the shell and scales with its thickness, differently from what happens in the post-buckling regime (after a dimple has formed). Consequently, in the pre-buckling regime, the change in volume scales as R^2h . A more formal way to obtain this is to compute the difference in volume assuming a normal displacement of order h so that the deformed radius is equal to $R-h$. In this case, purely from geometry, the change in volume is

$$\Delta V = \frac{4}{3}\pi (R-h)^3 - \frac{4}{3}\pi R^3 = 4\pi R^2 h \left(1 + \frac{h}{R} + \frac{1}{3}\left(\frac{h}{R}\right)^2\right) \sim R^2 h$$

Given that we have chosen to nondimensionalize the volume by R^2h , the dimensionless change in volume will be of order 1.”

The rest of the analysis consists only in the comparison of the several energy terms, which leads to the final scaling for the change in knockdown factor in Eq. (S58).

(3) The authors show that the change in knock-down factor induced by changing the parameter Λ_m , but is more or less independent of the defect amplitude, δ . This seems very surprising, but I could not really understand from the analysis presented why this should be the case - can the authors offer any physical intuition about this?

This is an important point. The fact that the *change* of knockdown factor is significantly less sensitive to defects than the knockdown factor itself highlights the robustness of the proposed mechanics. This result can be interpreted by the mechanism of tunability: the change of knockdown factor is dictated by the magnetic torque generated due to the shell rotation, which breaks the alignment between the shell magnetization and the applied field. From this effect, we can see that the generation of the torque is not constrained to the amplitude and width of the defect. Therefore, the change of knockdown factor should be insensitive to the defect ($\Delta\kappa_d$ in Fig. 3b). However, we still observed a slight dependence on the defect amplitude for $\delta/h < 1$. This might be because, the magnetic effect (including the tangential and normal forces, in addition to the magnetic torque, as detailed in Supplementary Note S3.2) slightly changes the defect geometry, which could result in a prominent change in $\Delta\kappa_d$ due to the imperfection sensitivity in the range $\delta/h < 1$. On the other hand, for $\delta/h > 1$, both the knockdown factor (κ_d in Fig. 3a) and its change ($\Delta\kappa_d$ in Fig. 3b) are not sensitive to the defect.

(4) It would be helpful if the authors could relate their work to the other bodies of work mentioned in (i) and (ii) above. For example, the authors’ results show that the knockdown parameter is proportional to the magnetic parameter Λ_m . It seems that this is almost the same as saying that the effective modulus of the shell is changed from the ‘true’ material E , to an effective value that supplements the modulus with an additional one proportional to the product of the residual and applied magnetic fields. Is this correct, and can it be understood qualitatively as the effect of a magnetism-induced elasticity, described in (ii) above?

Above, in our response to Comment 3.2, we have detailed the mechanism of the work by Vella et al. [R3.16] and Schönke & Fried [R3.17]. This chain of hard-ferromagnetic spheres is a pure magnetic system, with an “effective stiffness” that stems from the strong internal particle interaction. By contrast, in our system, the magnetic effect arising from the interaction between the magnetization of the shell and the external field is the ingredient responsible for modifying the knockdown factor via the magneto-elastic coupling. Specifically, a distributed magnetic torque is

generated and imposed on the shell due to its rotation prior to buckling, which gradually increases with the shell rotation and is localized near the region where buckling occurs (see Fig. S5 in Supplementary Note S3.4). This interpretation was corroborated by the theoretical model that we developed, which has been thoroughly validated by the experiments. This mechanism demonstrates that the magneto-elastic coupling is not merely equivalent to a change in the “effective modulus”. Further evidence to this interpretation is provided by the loading curves shown in Fig. 1d, where the slope in the linear regime (stiffness of the shell) does not change at different levels of field strength, but the value of pressure at which buckling occurs and its drop in the post-buckling regimes varies significantly with the applied field. This indicates our magnetic shells become stronger/weaker past the critical point, and the magnetic field is able to qualitatively change the nature of the buckling event, making it less/more catastrophic; an effect that cannot be attributed by a simple change in an effective modulus.

As another important related point, the second-order term of the magnetic energy (Eq. (S45) of the SI), which dominates the change of knockdown factor, is related to the shell rotation (\mathbf{q} defined in Eq. (S44) of the SI). On the other hand, the elastic energy (Eq. (S21) of the SI) depends on the stretching deformation (measured by the membrane strain tensor $\mathbf{E} = (\mathbf{a} - \hat{\mathbf{a}})/2$) and the bending deformation (measured by the curvature tensor $\mathbf{b} - \hat{\mathbf{b}}$) of the shell. This difference in nature between the magnetic and elastic energies further indicates that the magnetic energy can not simply be considered as an “additional elastic energy” through an “effective modulus”.

In summary, although the rotational stiffness defined in Eq. (S46) of our work and the bending stiffness of magnetic chains in Refs. [R3.16,R3.17] are both proportional to B^2/μ_0 , the underlying mechanism is fundamentally different.

(5) As far as I could tell, the residual field is always applied in the direction opposing the action of the pressure. I think that flipping the sign of \mathbf{B}^r does not change anything, since it is only the sign of the product $\mathbf{B}^r \cdot \mathbf{B}^a$ that matters, but it might be worth noting that this is the case.

We fully agree with the reviewer; the sign of the dot product $\mathbf{B}^r \cdot \mathbf{B}^a$ determines the direction of the torque and the sign of $\Delta\kappa_d$. We have noted this in the revised manuscript when discussing the mechanism. We did not provide this statement in the original version of the manuscript since we were only considering the axisymmetric cases (\mathbf{B}^r and \mathbf{B}^a are aligned), while the dot product $\mathbf{B}^r \cdot \mathbf{B}^a$ also includes asymmetric cases. Following a suggestion by another Reviewer, we have performed extensive experiments and FEM simulations to explore non-axisymmetric cases (where the defect is not aligned with the magnetization \mathbf{B}^r and/or the applied field \mathbf{B}^a). The results show that the proposed mechanism is still preserved. Since this really extends our conclusion and strengthens this work, we would like to summarize the results in the following, which has also been added to the main article in Discussion.

“Thus far, we have demonstrated the possibility of tuning the knockdown factor of axisymmetric shells via magnetic actuation. This change of knockdown factor is dictated by the magnetic torque generated due to the shell rotation, which breaks the alignment between the magnetization (\mathbf{B}^r) and the applied field (\mathbf{B}^a). Consequently, this mechanism is expected to be preserved for asymmetric loading conditions, provided \mathbf{B}^r and \mathbf{B}^a are aligned in the initial configuration. To attest this statement, we first magnetize our shell specimens asymmetrically, at an angle φ^r with respect to the shell’s axis of symmetry (see schematic diagram in Fig. R3.2a). The defect is still maintained at the pole. Then, during the buckling test, a magnetic field at the same angle ($\varphi^a = \varphi^r$) is applied, either parallel ($\mathbf{B}^r \cdot \mathbf{B}^a > 0$) or antiparallel ($\mathbf{B}^r \cdot \mathbf{B}^a < 0$) to the initial shell

magnetization. In Fig. R3.2a, we plot the measured change of knockdown factor, $\Delta\kappa_d$, versus φ^r and φ^a , at $\Lambda_m = \pm 0.22$. We consistently observe finite values of $\Delta\kappa_d$ for both the axisymmetric and asymmetric cases, even when \mathbf{B}^r and \mathbf{B}^a are perpendicular to the axis of the defect. In the latter, the most unfavourable case, the range of ± 0.1 in the tuning of $\Delta\kappa_d$ is still significant. These results indicate that, even when the defect information might be unknown in an application setting, the tunable mechanism that we uncovered can always be realized by aligning the applied field with the magnetization. Still, at the same level of Λ_m , the capacity of tuning decreases with increasing φ^r and φ^a , and the axisymmetric configuration is the most efficient at maximizing $\Delta\kappa_d$.

To further probe the robustness of the proposed mechanism, we consider asymmetric cases, where \mathbf{B}^r and \mathbf{B}^a were misaligned. We focus on shell specimens with a fixed magnetization set at $\varphi^r = 0^\circ$, while the direction of the applied field is varied by increasing φ^a from 0° (axisymmetric, \mathbf{B}^a is parallel to \mathbf{B}^r) to 90° (\mathbf{B}^a is perpendicular to \mathbf{B}^r). The corresponding change of knockdown factor is plotted in Fig. R3.2b, which (the absolute value) decreases from the maximum at $\varphi^a = 0^\circ$ to $\Delta\kappa_d = 0$ at $\varphi^a = 90^\circ$. Thus, in settings where the alignment between the shell magnetization and the external field cannot be ensured, tuning the knockdown factor is always possible (unless \mathbf{B}^a is perpendicular to \mathbf{B}^r , which leads to $\Delta\kappa_d = 0$). This relaxation in loading conditions broadens the applicability of the mechanism in real applications involving complex magnetization profiles and magnetic fields. In parallel to the experiments, we perform full 3D FEM simulations (see Methods for details) using a user-defined element proposed in Ref. [R3.6] for hard-magnetic soft materials. Good agreement between FEM and experiments is found in Fig. R3.2a,b, which further corroborates our findings."

Fig. R3.2: **Tunability under asymmetric loading conditions.** **a**, Change of knockdown factor, $\Delta\kappa_d$, at $\Lambda_m = 0.22$ ($\mathbf{B}^r \cdot \mathbf{B}^a > 0$) and $\Lambda_m = -0.22$ ($\mathbf{B}^r \cdot \mathbf{B}^a < 0$), versus angle of the shell magnetization, φ^r , and of the applied field, φ^a , which are consistently aligned in the initial configuration ($\varphi^a = \varphi^r$). In the experiments, 16 shell specimens containing a defect at the pole with identical geometry are fabricated (with average defect amplitude $\delta/h = 1.64 \pm 0.04$ and defect width $\varphi_0 = 11.7^\circ$). The shells are magnetized at an angle φ^r with respect to the axis of symmetry (2 or 3 specimens at each angle, as indicated by the number of data points). Each specimen (magnetized at φ^r) is tested with the magnetic field oriented at $\varphi^a = \varphi^r$. **b**, Change of knockdown factor, $\Delta\kappa_d$, at $\Lambda_m = 0.22$ ($\mathbf{B}^r \cdot \mathbf{B}^a > 0$) and $\Lambda_m = -0.22$ ($\mathbf{B}^r \cdot \mathbf{B}^a < 0$), as a function of φ^a , with fixed $\varphi^r = 0^\circ$. The two shell specimens used for the data in panel **b** were the same as those from panel **a** that were magnetized at $\varphi^r = 0^\circ$.

3.3 Minor comments:

- (1) In figure S2 I think the geometrical parameters of the shell (R and h) should be given too, since this is important in determining the critical pressure.

As suggested, in the caption of Fig. S2, we have included the values of the radius and thickness, which are the same as the geometric information given in the “Methods (Geometry of the shells)” section.

- (2) After eqn (S55), it would be helpful to write $\kappa_d \sim (p/E)(R/h)^2$ (brackets added) to be clear about whether the $(R/h)^2$ should be on the denominator or not.

As suggested, we have added parentheses to make the expression clear, in the relevant places, both in the main text and the SI.

- (3) Page 16 of the SI: “compassion” should be “comparison”.

Thanks for pointing out the typo, which has been fixed.

References used in our responses to the Reviewer’s comments/questions:

- [R3.1] Tsien, H.-S. A theory for the buckling of thin shells. *Journal of the Aeronautical Sciences* **9**, 373-384 (1942).
- [R3.2] Hutchinson, J. W. Imperfection sensitivity of externally pressurized spherical shells. *J. Appl. Mech.* **34**, 49-55 (1967).
- [R3.3] Carlson, R. L., Sendelbeck, R. L. & Hoff, N. J. Experimental studies of the buckling of complete spherical shells. *Exp. Mech.* **7**, 281-288 (1967).
- [R3.4] Lum, G. Z. et al. Shape-programmable magnetic soft matter. *Proc. Natl. Acad. Sci. U.S.A.* **113**, E6007-E6015 (2016).
- [R3.5] Kim, Y., Yuk, H., Zhao, R., Chester, S. A. & Zhao, X. Printing ferromagnetic domains for untethered fast-transforming soft materials. *Nature* **558**, 274 (2018).
- [R3.6] Zhao, R., Kim, Y., Chester, S. A., Sharma, P. & Zhao, X. Mechanics of hard-magnetic soft materials. *J. Mech. Phys. Solids* **124**, 244-263 (2019).
- [R3.7] Wang, L., Kim, Y., Guo, C. F. & Zhao, X. Hard-magnetic elastica. *J. Mech. Phys. Solids* **142**, 104045 (2020).
- [R3.8] Gorissen, B., Melancon, D., Vasios N., Torbati, M. & Bertoldi, K. Inflatable soft jumper inspired by shell snapping. *Science Robotics* **5**, eabb1967 (2020).
- [R3.9] Moon, F. C. & Pao, Y.-H. Magnetoelastic buckling of a thin plate. *J. Appl. Mech.* **35**, 53-58 (1968).
- [R3.10] Cēbers, A. Dynamics of a chain of magnetic particles connected with elastic linkers. *J. Phys.: Condens. Matter* **15**, S1335-S1344 (2003).
- [R3.11] Cēbers, A. & Javaitis I. Bending of flexible magnetic rods. *Phys. Rev. E* **70**, 021404 (2004).
- [R3.12] Dreyfus, R., Baudry, J., Roper, M. L., Fermigier, M., Stone, H. A. & Bibette, J. Microscopic artificial swimmers. *Nature* **437**, 862-865 (2005).
- [R3.13] Roper, M., Dreyfus, R., Baudry, J., Fermigier, M., Bibette, J. & Stone, H. A. On the dynamics of magnetically driven elastic filaments. *J. Fluid Mech.* **554**, 167-190 (2006).
- [R3.14] Cēbers, A. & Ćirulis, T. Magnetic elastica. *Phys. Rev. E* **76**, 031504 (2007).
- [R3.15] Gerbal, F., Wang, Y., Lyonnet, F., Bacri, J.-C., Hocquet, T. & Devaud, M. A refined theory of magnetoelastic buckling matches experiments with ferromagnetic and superparamagnetic rods. *Proc. Natl. Acad. Sci. U.S.A.* **112**, 7135-7140 (2015).
- [R3.16] Vella, D., du Pontavice, E., Hall, C. L. & Goriely, A. The magneto-elastica: from self-buckling to self-assembly. *Proc. R. Soc. A* **470**, 20130609 (2014).

- [R3.17] Schönke, J & Fried, E. Stability of vertical magnetic chains. *Proc. R. Soc. A* **473**, 20160703 (2017).
- [R3.18] Rigbi, Z. & Jilken, L. The response of an elastomer filled with soft ferrite to mechanical and magnetic influences. *J. Magn. Magn. Mater.* **37**, 267–276 (1983).
- [R3.19] Ginder, J., Nichols, M., Elie, L. & Tardiff, J. Magnetorheological elastomers: properties and applications. In: M. Wuttig (Ed.), *Smart Structures and Materials 1999*, Proceedings of the SPIE, vol. 3675, Smart Materials Technologies, 131–138 (1999).
- [R3.20] Dorfmann, A. & Ogden, R. Magnetoelastic modelling of elastomers. *Eur. J. Mech. A/Solids* **22**, 497-507 (2003).
- [R3.21] Danas, K., Kankanala, S. V. & Triantafyllidis, N. Experiments and modeling of iron-particle-filled magnetorheological elastomers. *J. Mech. Phys. Solids* **60**, 120-138 (2012).
- [R3.22] Loukaides, E. G., Smoukov, S. K. & Seffen, K. A. Magnetic actuation and transition shapes of a bistable spherical cap. *Int. J. Smart Nano. Mater.* **5**, 270-282 (2014).
- [R3.23] Seffen, K. A. & Vidoli, S. Eversion of bistable shells under magnetic actuation: a model of nonlinear shapes. *Smart Mater. Struct.* **25**, 065010 (2016).
- [R3.24] Psarra, E., Bodelot, L. & Danas, K. Wrinkling to crinkling transitions and curvature localization in a magnetoelastic lm bonded to a non-magnetic substrate. *J. Mech. Phys. Solids* **133**, 103734 (2019).
- [R3.25] Hu, W., Lum, G. Z., Mastrangeli, M. & Sitti, M. Small-scale soft-bodied robot with multimodal locomotion. *Nature* **554**, 81-85 (2018).
- [R3.26] Kim, Y., Parada, G. A., Liu, S. & Zhao, X. Ferromagnetic soft continuum robots. *Sci. Robot.* **4**, eaax7329 (2019).
- [R3.27] Gu, H. et al. Magnetic cilia carpets with programmable metachronal waves. *Nat. Commun.* **11**, 2637 (2020).
- [R3.28] Alapan, Y., Karacakol, A. C., Guzelhan, S. N., Isik, I. & Sitti, M. Reprogrammable shape morphing of magnetic soft machines. *Sci. Adv.* **6**, eabc6414 (2020).

Reviewers' Comments:

Reviewer #1:

Remarks to the Author:

Thank you for your detailed response to my comments. I have nothing further to add concerning the paper.

Reviewer #2:

Remarks to the Author:

Thank you for addressing the comments comprehensively. I am very happy with this paper, and I think it should do well out there having an impact on the structural-elasticity community.

Reviewer #3:

Remarks to the Author:

The authors have made a number of changes to the manuscript in response to my comments and questions. I have two comments regarding their responses:

Firstly, regarding the neglect of magnetic interactions between embedded particles themselves: I find the new experiments the authors present in figure R3.1 very convincing. It would still be nice to understand why this is the case: since B_r and B_a are of similar magnitude (e.g. around 60 mT in figure 1) the interaction of the remanant magnetization of one particle with that of another nearby would (I think) be expected to be order $|B_r|^2$, while the energy of the interaction with the applied field is order $B_r B_a$. Since these two are of similar size experimentally, it seems at first sight that their energies should also be of the same order. I guess the authors are right that it is the low concentration that is missing in the argument above, but a note to explain why in the manuscript and/or a copy of Fig R3.1 in the supplementary information would be helpful to others who might ask a similar question.

Secondly, regarding the computation of the change in volume, I am now even more confused: the authors note that the change in volume before buckling ($\Delta V_{pre} \sim h R^2$) is larger than that post-buckling (which they agree is $\Delta V_{post} \sim h^2 R$). (Since the aspect ratio h/R is small, it is clear that if these scalings really are correct then $\Delta V_{pre}/\Delta V_{post} \sim R/h \gg 1$.) If this is really what they mean, I think it would deserve some discussion/explanation around eqn (S.53).

Response to the Reviewers' Comments for Manuscript NCOMMS-20-32420A
"Magneto-active elastic shells with tunable buckling strength"
Dong Yan, Matteo Pezulla, Lilian Cruveiller, Arefeh Abbasi, Pedro M. Reis

Please refer to the revised version of our manuscript for a detailed account of all the changes, corrections and additions; the revised/new text is **marked in blue**. In the following, we address each of the Reviewers' points (**original comments in bold**). The numbering of the comments (e.g., 1.1, 1.2, ..., 2.1, 2.2,...) is added by us to aid in cross-referencing our responses. A list of relevant references, some of which are not included in the manuscript, is provided (e.g., as [RR1.1], [RR1.2],..., [RR2.1], ...) at the end of the response to each Reviewer.

Summary of changes:

- We have added a clear statement that the shell model that we have developed does not consider the effect of magnetic self-interactions, providing the supporting experimental evidence in *Supplementary Note S3.1*.
- Experimental evidence to the point above was added in a new figure (Fig. S4), which includes two plots.
- We have added a new paragraph to *Supplementary Note S3.3* to discuss the scalings of the change of the shell's volume across the pre- and post-buckling regimes. Moreover, in this new paragraph, we highlight that the scaling we use to predict the critical buckling conditions corresponds to a uniformly compressed sphere.

1. Response to Reviewer #1

Thank you for your detailed response to my comments. I have nothing further to add concerning the paper.

We are glad to hear that the Reviewer found our thorough response satisfactory and complete.

2. Response to Reviewer #2

Thank you for addressing the comments comprehensively. I am very happy with this paper, and I think it should do well out there having an impact on the structural-elasticity community.

We appreciate the Reviewer's encouraging comment on the potential impact of our paper.

3. Response to Reviewer #3

The authors have made a number of changes to the manuscript in response to my comments and questions. I have two comments regarding their responses:

We very much appreciate the Reviewer's dedication, as well as the additional questions and suggestions, all of which have been implemented to improve the paper.

3.1 Firstly, regarding the neglect of magnetic interactions between embedded particles themselves: I find the new experiments the authors present in figure R3.1 very convincing. It would still be nice to understand why this is the case: since B_r and B_a are of similar magnitude (e.g. around 60 mT in figure 1) the interaction of the remanant magnetization of one particle with that of another nearby would (I think) be expected to be order $|B_r|^2$, while the energy of the interaction with the applied field is order $B_r B_a$. Since these two are of similar size experimentally, it seems at first sight that their energies should also be of the same order. I guess the authors are right that it is the low concentration that is missing in the argument above, but a note to explain why in the manuscript and/or a copy of Fig R3.1 in the supplementary information would be helpful to others who might ask a similar question.

We appreciate the Reviewer's positive feedback on the additional experiments that we had performed following the previous request, which allowed us to more thoroughly evaluate the influence of particle interactions on our magnetic shell system. We agree with the Reviewer that it is appropriate to include these results in the manuscript, which we now do in its new revised version (in the Supplementary Notes). Specifically, we have added the following note to the Section "**Theory of axisymmetric hard-magnetic shells**" of the main article:

"In this model, the magnetic interaction between particles embedded in the MRE is not taken into account due to its negligible influence on the results, which has been validated through experiments (evidence provided in Supplementary Note S3)."

Concurrently, in *Supplementary Note S3.1*, we present the new experimental results (Fig. S4, with the accompanying discussion), which provide the justifying evidence for the approximation stated above. The following text was added to the paragraph where the expression of magnetic energy, Eq. (S4), is introduced:

“In this model, we have neglected the energy corresponding to the magnetic interaction between particles embedded in the MRE. To test the validity of this assumption, we performed a specific test of experiments that confirmed these magnetic interactions do not modify the original shell geometry, nor do they affect the knockdown factor. In the experiments, differently from the fabrication protocol described in Section S1.1, we peeled off each of the shells (Fig. S1e) from the mold before they were magnetized. We then quantified the defect amplitude of each non-magnetized shell and measured their critical buckling pressure. The shells were then magnetized, after which the same measurements were performed (defect amplitude and critical buckling pressure). No external magnetic field was applied during these buckling tests to evaluate the influence of particle interactions alone. In Fig. S4, we present the experimental results for a set of eight, across a range of small ($\bar{\delta} < 1$) and large ($\bar{\delta} > 1$) defects. We found that the defect amplitudes (Fig. S4a) and the corresponding knockdown factors (Fig. S4b) measured before and after magnetizing were nearly identical. These results indicate that the particle-particle magnetic interactions were too weak to affect the shells’ geometry and buckling behavior, presumably due to the low volume concentration and random distribution of the magnetic particles in the polymer matrix of our MRE. Also, before the onset of buckling, different regions in the shell are still distanced sufficiently; thus, minimizing the associated self-interaction. For more complex settings, where particle interactions may play a role, a higher level of sophistication in the modeling approach will be required, beyond the scope of our present study.”

Fig. S4: Evaluation of the potential effect of particle-particle magnetic interactions. We tested eight different shell specimens with different degrees of imperfection. **a**, Normalized defect amplitudes, $\bar{\delta}$, and **b**, the corresponding knockdown factors, κ_d , of shell specimens before (circles) and after (crosses) they were magnetized. The two sets of measurements are nearly identical, indicating that magnetic interactions are negligible.

Just like the Reviewer, we would have loved to provide a strong theoretical argument to rationalize the above results but, in all honesty, we have not been able to devise a formal explanation on this point given the complexity of the coupled-field mechanism at play. Still, in the following, we seek to provide an interpretation that we believe may shed some physical insight into this issue.

We agree that the energy corresponding to particle-particle magnetic interactions is proportional to $\mu_0^{-1} \|\mathbf{B}^r\|^2$, which is of the same order as $\mu_0^{-1} \|\mathbf{B}^a\| \|\mathbf{B}^r\|$, but the scale of this energy is also expected to depend on other quantities dictated by the shell deformation (changes of the distances and angles between particles). For example, the expressions for the various orders and terms of magnetic energy corresponding to the interaction between the MRE and the external field — the different terms in Eq. (S51)-(S53) (Supplementary Note S3.3) — are all proportional to $\mu_0^{-1} \|\mathbf{B}^a\| \|\mathbf{B}^r\|$; however, the power of (h/R) determines the specifics of their different scales. As such, to perform a direct comparison between the energy of the interaction between the external field and particles (considered in our model) and the energy of particle-particle magnetic interactions (not considered in our model), we would have needed to derive complete expressions for the scaling of the latter. Unfortunately, this is a non-trivial task beyond our expertise on the modeling of MREs. We hope that the experimental evidence provided in the data presented in Fig. S4 will instigate future theoretical work in this direction, as well as motivate new experimental investigations under conditions where self-interactions could play a functional role.

3.2 Secondly, regarding the computation of the change in volume, I am now even more confused: the authors note that the change in volume before buckling ($\Delta V_{\text{pre}} \sim h R^2$) is larger than that post-buckling (which they agree is $\Delta V_{\text{post}} \sim h^2 R$). (Since the aspect ratio h/R is small, it is clear that if these scalings really are correct then $\Delta V_{\text{pre}}/\Delta V_{\text{post}} \sim R/h \gg 1$.) If this is really what they mean, I think it would deserve some discussion/explanation around eqn (S.53).

The scaling of the change in volume in post-buckling, denoted by ΔV_{post} , is valid only in the early stages of the post-buckling regime. This statement can be rationalized by the fact that the normal displacement is assumed to be of the order of the shell thickness, which is valid only up to and just past the onset of buckling (*i.e.*, in the linear regime of deformation). Deeper into the postbuckling regime, when the dimple is clearly visible, the normal displacement is as large as ten times the thickness. Consequently, the scaling for ΔV_{post} is no longer valid in this regime given that the deformation becomes geometrically nonlinear.

In Fig. RR3.1, we plot a pressure-volume curve computed using our theoretical model for a shell with radius $R=25.4$ mm and thickness $h=0.32$ mm. Along the loading curves, in the insets, we provide the shape of the shell, marking whether it corresponds to a stable state (closed symbol) or an unstable state (open symbol). Note that a small geometric imperfection (amplitude $\delta/h=0.001$, half angular width $\varphi_0=1^\circ$) has been seeded into the numerics to ease computation, resulting in a critical load that is slightly smaller than the theoretical one; but this (arte)fact does not affect the discussion below.

Right after buckling (point with $w_{\text{pole}}/h=1$ in Fig. RR3.1), the shell is located in the unstable branch (dashed line) of the loading path; therefore, it is not observable under imposed-volume conditions (as we do in our experiments). As stated previously, past the buckling event, the scaling for ΔV_{post} only holds when the normal displacement is of the order of the thickness of the shell, which we now see occurs in the unstable portion of the loading curve. Hence, when comparing ΔV_{pre} and ΔV_{post} , it is clear, as the Reviewer states, that $\Delta V_{\text{pre}} > \Delta V_{\text{post}}$. It is also worth noting that, depending on the boundary conditions, the prefactors for the scalings of these two volumes might differ; something that we do not know how to compute analytically for ΔV_{post} . The above statement that $\Delta V_{\text{pre}} > \Delta V_{\text{post}}$ may come across as counterintuitive, but can be understood by focusing on the pressure-volume loading curve in Fig.

RR3.1. Right past buckling, the unstable equilibrium branch closely follows the pre-buckling (stable) patch, with a decreasing ΔV . Eventually, further down in the postbuckling regime, for displacements considerably larger than the thickness, the variation in volume will overshoot the variation in volume at buckling (e.g., the $w_{\text{pole}}/h=13.4$ case) and the $\Delta V_{\text{pre}} > \Delta V_{\text{post}}$ is no longer true. However, the cases deeper into the postbuckling regime are not studied in the manuscript since our focus is primarily on the quantification of the knockdown factor (definition determined at the onset of buckling).

Fig. RR3.1 Loading curve of pressure, p/p_c , versus volume change, $\Delta V/\Delta V_c$ for a perfect shell computed from our shell model. A small geometric imperfection (amplitude $\delta/h=0.001$, half angular width $\varphi_0=1^\circ$) has been seeded into the numerics to ease computation, resulting in a critical load that is slightly smaller than the theoretical one. Insets: The shapes of the deformed shell along different points on the loading curve are presented with the corresponding values of the deflections at the pole (normalized by shell thickness, w_{pole}/h). In the $\bar{p}(\Delta\bar{V})$ curves, the stable states are marked with closed circles and the unstable states with open circles. The radius and thickness of this shell are, respectively, $R=25.4$ mm and $h=0.32$ mm, yielding $R/h = 79$. The classic critical buckling pressure and the corresponding volume change immediately prior to buckling are given by $p_c=2E[3(1-\nu^2)]^{-1/2}(R/h)^{-2}$ and $\Delta V_c=2\pi(1-\nu)[3(1-\nu^2)]^{-1/2}(R^2h)$, respectively [RR3.1].

Motivated by the Reviewer's comment, we have added the following sentences in Supplementary Note S3.3 (in the subsection on the scaling of the potential associated to the live pressure):

"We highlight that the above scaling argument for the critical volume change, ΔV , is based on a uniformly compressed sphere, corresponding to deformations where a dimple has not yet formed. Indeed, right past the buckling event (i.e., when a dimple has formed), the analogous scaling to Eq. (S54) would be $\Delta V \sim h^2R$, given that the height of the dimple is assumed to be of the order of the thickness and its width scales as \sqrt{Rh} . This scaling argument is only valid immediately after buckling, that is, for unstable shapes, and cannot be used in general for the postbuckling regime as a whole, where the amplitude of the dimple becomes of the same order of shell radius."

References used in the present response:

[RR3.1] Hutchinson, J. W. & Thompson, J. M. T. Nonlinear buckling behaviour of spherical shells: barriers and symmetry-breaking dimples. *Phil. Trans. R. Soc. A* **375**, 20160154 (2017).